# Sliding Window Algorithms for k-Clustering Problems

**Michele Borassi**
Google Zürich
borassi@google.com

**Alessandro Epasto**
Google Research New York
aepasto@google.com

**Silvio Lattanzi**
Google Research Zürich
silviol@google.com

**Sergei Vassilviskii**
Google Research New York
sergeiv@google.com

**Morteza Zadimoghaddam**
Google Research Cambridge
zadim@google.com

## Abstract

The sliding window model of computation captures scenarios in which data is arriving continuously, but only the latest $w$ elements should be used for analysis. The goal is to design algorithms that update the solution efficiently with each arrival rather than recomputing it from scratch. In this work, we focus on $k$-clustering problems such as $k$-means and $k$-median. In this setting, we provide simple and practical algorithms that offer stronger performance guarantees than previous results. Empirically, we show that our methods store only a small fraction of the data, are orders of magnitude faster, and find solutions with costs only slightly higher than those returned by algorithms with access to the full dataset.

## 1 Introduction

Data clustering is a central tenet of unsupervised machine learning. One version of the problem can be phrased as grouping data into $k$ clusters so that elements within the same cluster are similar to each other. Classic formulations of this question include the $k$-median and $k$-means problems for which good approximation algorithms are known [1, 44]. Unfortunately, these algorithms often do not scale to large modern datasets requiring researchers to turn to parallel [8], distributed [9], and streaming methods. In the latter model, points arrive one at a time and the goal is to find algorithms that quickly update a small *sketch* (or summary) of the input data that can then be used to compute an approximately optimal solution.

One significant limitation of the classic data stream model is that it ignores the time when a data point arrived; in fact, all of the points in the input are treated with equal significance. However, in practice, it is often important (and sometimes necessary) to restrict the computation to very recent data. This restriction may be due to data freshness—e.g., when training a model on recent events, data from many days ago may be less relevant compared to data from the previous hour. Another motivation arises from legal reasons, e.g., data privacy laws such as the General Data Protection Regulation (GDPR), encourage and mandate that companies not retain certain user data beyond a specified period. This has resulted in many products including a data retention policy [54]. Such recency requirements can be modeled by the *sliding window* model. Here the goal is to maintain a small sketch of the input data, just as with the streaming model, and then use only this sketch to approximate the solution on the last $w$ elements of the stream.

Clustering in the sliding window model is the main question that we study in this work. A trivial solution simply maintains the $w$ elements in the window and recomputes the clusters from scratch at each step. We intend to find solutions that use less space, and are more efficient at processing each

new element. In particular, we present an algorithm which uses space linear in $k$, and polylogarithmic in $w$, but still attains a constant factor approximation.

**Related Work**  *Clustering.* Clustering is a fundamental problem in unsupervised machine learning and has application in a disparate variety of settings, including data summarization, exploratory data analysis, matrix approximations and outlier detection [39, 41, 46, 50]. One of the most studied formulations in clustering of metric spaces is that of finding $k$ centers that minimize an objective consisting of the $\ell_p$ norm of the distances of all points to their closest center. For $p \in \{1, 2, \infty\}$ this problem corresponds to $k$-median, $k$-means, and $k$-center, respectively, which are NP-hard, but constant factor approximation algorithms are known [1, 34, 44]. Several techniques have been used to tackle these problems at scale, including dimensionality reduction [45], core-sets [6], distributed algorithms [5], and streaming methods reviewed later. To clarify between Euclidean or general metric spaces, we note that our results work on arbitrary general metric spaces. The hardness results in the literature hold even for special case of Euclidean metrics and the constant factor approximation algorithms hold for the general metric spaces.

*Streaming model.* Significant attention has been devoted to models for analyzing large-scale datasets that evolve over time. The streaming model of computation is of the most well-known (see [49] for a survey) and focuses on defining low-memory algorithms for processing data arriving one item at a time. A number of interesting results are known in this model ranging from the estimation of stream statistics [3, 10], to submodular optimization [7], to graph problems [2, 30, 42], and many others. Clustering is also well studied in this setting, including algorithms for $k$-median, $k$-means, and $k$-center in the insertion-only stream case [6, 20, 35].

*Sliding window streaming model.* The sliding window model significantly increases the difficultly of the problem, since deletions need to be handled as well. Several techniques are known, including the exponential histogram framework [27] that addresses weakly additive set functions, and the smooth histogram framework [18] that is suited for functions that are well-behaved and possesses a sufficiently small constant approximation. Since many problems, such as $k$-clustering, do not fit into these two categories, a number of algorithms have been developed for specific problems such as submodular optimization [14, 21, 29], graph sparsification [26], minimizing the enclosing ball [55], private heavy hitters [54], diversity maximization [14] and linear algebra operations [15]. Sliding window algorithms find also applications in data summarization [23].

Turning to sliding window algorithms for clustering, for the $k$-center problem Cohen et al. [25] show a $(6 + \epsilon)$-approximation using $O(k \log \Delta)$ space and per point update time of $O(k^2 \log \Delta)$, where $\Delta$ is the spread of the metric, i.e. the ratio of the largest to the smallest pairwise distances. For $k$-median and $k$-means, [17] give constant factor approximation algorithms that use $O(k^3 \log^6 w)$ space and per point update time of $O(poly(k, \log w))$.[1] Their bound is polylogarithmic in $w$, but *cubic* in $k$, making it impractical unless $k \ll w$.[2] In this paper we improve their bounds and give a simpler algorithm with only linear dependency of $k$. Furthermore we show experimentally (Figure 1 and Table 1) that our algorithm is faster and uses significantly less memory than the one presented in [17] even with very small values $k$ (i.e., $k \geq 4$). In a different approach, [56] study a variant where one receives points in batches and uses heuristics to reduce the space and time. Their approach does provide approximation guarantees but it applies only to the Euclidean k-means case. Recently, [32] studied clustering problems in the distributed sliding window model, but these results are not applicable to our setting.

The more challenging fully-dynamic stream case has also received attention [16, 38]. Contrary to our result for the sliding window case, in the fully-dynamic case, obtaining a $\tilde{\mathcal{O}}(k)$ memory, low update time algorithm, for the *arbitrary* metric $k$-clustering case with general $\ell_p$ norms is an open problem. For the special case of $d$-dimensional Euclidean spaces for $k$-means, there are positive results—[38] give $\tilde{O}(kd^4)$-space core-set with $1 + \epsilon$ approximation.

Dynamic algorithms have also been studied in a consistent model [24, 43], but there the objective is to minimize the number of changes to the solution as the input evolves, rather than minimizing the approximation ratio and space used. Finally, a relaxation of the fully dynamic model that allows only

a limited number of deletions has also been addressed [33, 48]. The only work related to clustering is that of submodular maximization [48] which includes exemplar-based clustering as a special case.

**Our Contributions** We simplify and improve the state-of-the-art of $k$-clustering sliding window algorithms, resulting in lower memory algorithms. Specifically, we:

- Introduce a simple new algorithm for $k$-clustering in the sliding window setting (Section 3.2). The algorithm is an example of a more general technique that we develop for minimization problems in this setting. (Section 3).
- Prove that the algorithm needs space linear in $k$ to obtain a constant approximate solution (Theorem 3.4), thus improving over the best previously known result which required $\Omega(k^3)$ space.
- Show empirically that the algorithm is orders of magnitude faster, more space efficient, and more accurate than previous solutions, even for small values of $k$ (Section 4).

## 2 Preliminaries

Let $X$ be a set of arbitrary points, and $d : X \times X \to \mathbb{R}$ be a distance function. We assume that $(X, d)$ is an arbitrary metric space, that is, $d$ is non-negative, symmetric, and satisfies the triangle inequality. For simplicity of exposition we will make a series of additional assumptions, in supplementary material, we explain how we can remove all these assumptions. We assume that the distances are normalized to lie between 1 and $\Delta$. We will also consider weighted instances of our problem where, in addition, we are given a function weight : $X \to \mathbb{Z}$ denoting the multiplicity of the point.

The $k$-clustering family of problems asks to find a set of $k$ cluster centers that minimizes a particular objective function. For a point $x$ and a set of points $Y = \{y_1, y_2, \ldots, y_m\}$, we let $d(x, Y) = \min_{y \in Y} d(x, y)$, and let $\mathsf{cl}(x, Y)$ be the point that realizes it, $\arg\min_{y \in Y} d(x, y)$. The cost of a set of centers $\mathcal{C}$ is: $f_p(X, \mathcal{C}) = \sum_{x \in X} d^p(x, \mathcal{C})$. Similarly for weighted instances, we have $f_p(X, \mathsf{weight}, \mathcal{C}) = \sum_{x \in X} \mathsf{weight}(x) d^p(x, \mathcal{C})$.

Note that for $p = 2$, this is precisely the $k$-MEDOIDS problem.[3] For $p = 1$, the above encodes the $k$-MEDIAN problem. When $p$ is clear from the context, we will drop the subscript. We also refer to the optimum cost for a particular instance $(X, d)$ as $\mathrm{OPT}_p(X)$, and the optimal clustering as $\mathcal{C}_p^*(X) = \{\mathsf{c}_1^*, \mathsf{c}_2^*, \ldots, \mathsf{c}_k^*\}$, shortening to $\mathcal{C}^*$ when clear from context. Throughout the paper, we assume that $p$ is a constant with $p \geq 1$.

While mapping a point to its nearest cluster is optimal, any map $\mu : X \to X$ will produce a valid clustering. In a slight abuse of notation we extend the definition of $f_p$ to say $f_p(X, \mu) = \sum_{x \in X} d(x, \mu(x))^p$.

In this work, we are interested in algorithms for sliding window problems, we refer to the window size as $w$ and to the set of elements in the active window as $W$, and we use $n$ for the size of the entire stream, typically $n \gg w$. We denote by $X_t$ the $t$-th element of the stream and by $X_{[a,b]}$ the subset of the stream from time $a$ to $b$ (both included). For simplicity of exposition, we assume that we have access to a lower bound $m$ and upper bound $M$ of the cost of the optimal solution in any sliding window.[4]

We use two tools repeatedly in our analysis. The first is the relaxed triangle inequality. For $p \geq 1$ and any $x, y, z \in X$, we have: $d(x, y)^p \leq 2^{p-1}(d(x, z)^p + d(z, y)^p)$. The second is the fact that the value of the optimum solution of a clustering problem does not change drastically if the points are shifted around by a small amount. This is captured by Lemma 2.1 which was first proved in [35]. For completeness we present its proof in the supplementary material.

**Lemma 2.1.** *Given a set of points $X = \{x_1, \ldots, x_n\}$ consider a multiset $Y = \{y_1, \ldots, y_n\}$ such that $\sum_i d^p(x_i, y_i) \leq \alpha \, \mathrm{OPT}_p(X)$, for a constant $\alpha$. Let $\mathcal{B}^*$ be the optimal $k$-clustering solution for $Y$. Then $f_p(X, \mathcal{B}^*) \in O((1 + \alpha)OPT_p(X))$.*

Given a set of points $X$, a mapping $\mu : X \to Y$, and a weighted instance defined by $(Y, \text{weight})$, we say that the weighted instance is *consistent* with $\mu$, if for all $y \in Y$, we have that $\text{weight}(y) = |\{x \in X| \ \mu(x) = y\}|$. We say it is $\epsilon$-*consistent* (for constant $\epsilon \geq 0$), if for all $y \in Y$, we have that $|\{x \in X \mid \mu(x) = y\}| \leq \text{weight}(y) \leq (1+\epsilon)|\{x \in X \mid \mu(x) = y\}|$.

Finally, we remark that the $k$-clustering problem is NP-hard, so our focus will be on finding efficient approximation algorithms. We say that we obtain an $\alpha$ approximation for a clustering problem if $f_p(X, \mathcal{C}) \leq \alpha \cdot \text{OPT}_p(X)$. The best-known approximation factor for all the problems that we consider are constant [1, 19, 36]. Additionally, since the algorithms work in arbitrary metric spaces, we measure update time in terms of distance function evaluations and use the number of points as space cost (all other costs are negligible).

## 3 Algorithm and Analysis

The starting point of our clustering is the development of efficient sketching technique that, given a stream of points, $X$, a mapping $\mu$, and a time, $\tau$, returns a weighted instance that is $\epsilon$-consistent with $\mu$ for the points inserted at or after $\tau$. To see why having such a sketch is useful, suppose $\mu$ has a cost a constant factor larger than the cost of the optimal solution. Then we could get an approximation to the sliding window problem by computing an approximately optimal clustering on the weighted instance (see Lemma 2.1).

To develop such a sketch, we begin by relaxing our goal by allowing our sketch to return a weighted instance that is $\epsilon$-consistent with $\mu$ for the *entire stream* $X$ as opposed to the substream starting at $X_\tau$. Although a single sketch with this property is not enough to obtain a good algorithm for the overall problem, we design a sliding window algorithm that builds multiple such sketches in parallel. We can show that it is enough to maintain a polylogarithmic number of carefully chosen sketches to guarantee that we can return a good approximation to the optimal solution in the active window.

In subsection 3.1 we describe how we construct a single efficient sketch. Then, in the subsection 3.2, we describe how we can combine different sketches to obtain a good approximation. All of the missing proofs of the lemmas and the pseudo-code for all the missing algorithms are presented in the supplementary material.

### 3.1 Augmented Meyerson Sketch

Our sketching technique builds upon previous clustering algorithms developed for the streaming model of computation. Among these, a powerful approach is the sketch introduced for facility location problems by Meyerson [47].

At its core, given an approximate lower bound to the value of the optimum solution, Meyerson's algorithm constructs a set $\mathcal{C}$ of size $O(k \log \Delta)$, known as a *sketch*, and a consistent weighted instance, such that, with constant probability, $f_p(X, \mathcal{C}) \in O(\text{OPT}_p(X))$. Given such a sketch, it is easy to both: amplify the success probability to be arbitrarily close to 1 by running multiple copies in parallel, and reduce the number of centers to $k$ by keeping track of the number of points assigned to each $\mathsf{c} \in \mathcal{C}$ and then clustering this weighted instance into $k$ groups.

What makes the sketch appealing in practice is its easy construction—each arriving point is added as a new center with some carefully chosen probability. If a new point does not make it as a center, it is assigned to the nearest existing center, and the latter's weight is incremented by 1.

Meyerson algorithm was initially designed for online problems, and then adapted to algorithms in the streaming computation model, where points arrive one at a time but are never deleted. To solve the sliding window problem naively, one can simply start a new sketch with every newly arriving point, but this is inefficient. To overcome these limitations we extend the Meyerson sketch. In particular, there are two challenges that we face in sliding window models:

1. The weight of each cluster is not monotonically increasing, as points that are assigned to the cluster time out and are dropped from the window.

2. The designated center of each cluster may itself expire and be removed from the window, requiring us to pick a new representative for the cluster.

Using some auxiliary bookkeeping we can augment the classic Meyerson sketch to return a weighted instance that is $\epsilon$-consistent with a mapping $\mu$ whose cost is a constant factor larger than the cost of the optimal solution for the entire stream $X$. More precisely,

**Lemma 3.1.** *Let $w$ be the size of the sliding window, $\epsilon \in (0,1)$ be a constant and $t$ the current time. Let $(X, \mathrm{d})$ be a metric space and fix $\gamma \in (0,1)$. The augmented Meyerson algorithm computes an im-plicit mapping $\mu : X \to \mathcal{C}$, and an $\epsilon$-consistent weighted instance $(\mathcal{C}, \widehat{\mathrm{weight}})$ for all substreams $X_{[\tau,t]}$ with $\tau \geq t - w$, such that, with probability $1 - \gamma$, we have: $|\mathcal{C}| \leq 2^{2p+8} k \log \gamma^{-1} \log \Delta$ and $f_p(X_{[\tau,t]}, \mathcal{C}) \leq 2^{2p+8} \operatorname{OPT}_p(X)$.*

*The algorithm uses space $O(k \log \gamma^{-1} \log \Delta \log(M/m)(\log M + \log w + \log \Delta))$ and stores the cost of the consistent mapping, $f(X, \mu)$, and allows a $1 + \epsilon$ approximation to the cost of the $\epsilon$-consistent mapping, denoted by $\widehat{f}(X_{[\tau,t]}, \mu)$. This is the $\epsilon$-consistent mapping that is computed by the augmented Meyerson algorithm. In section 2, $M$ and $m$ are defined as the upper and lower bounds on the cost of the optimal solution.*

Note that when $M/m$ and $\Delta$ are polynomial in $w$,[5] the above space bound is $O(k \log \gamma^{-1} \log^3(w))$.

## 3.2 Sliding Window Algorithm

In the previous section we have shown that we can the Meyerson sketch to have enough information to output a solution using the points in the active window whose cost is comparable to the cost of the optimal computed on the whole stream. However, we need an algorithm that is competitive with the cost of the optimum solution computed solely on the elements in the sliding window.

We give some intuition behind our algorithm before delving into the details. Suppose we had a good guess on the value of the optimum solution, $\lambda^*$ and imagine splitting the input $x_1, x_2, \ldots, x_t$ into blocks $A_1 = \{x_1, x_2, \ldots, x_{b_1}\}$, $A_2 = \{x_{b_1+1}, \ldots, x_{b_2}\}$, etc. with the constraints that (i) each block has optimum cost smaller than $\lambda^*$, and (ii) is also maximal, that is adding the next element to the block causes its cost to exceed $\lambda^*$. It is easy to see, that any sliding window of optimal solution of cost $\lambda^*$ overlaps at most two blocks. The idea behind our algorithm is that, if we started an augmented Meyerson sketch in each block, and we obtain a good mapping for the suffix of the first of these two blocks, we can recover a good approximate solution for the sliding window.

We now show how to formalize this idea. During the execution of the algorithm, we first discretize the possible values of the optimum solution, and run a set of sketches for each value of $\lambda$. Specifically, for each guess $\lambda$, we run Algorithm 1 to compute the AugmentedMeyerson for two consecutive substreams, $A_\lambda$ and $B_\lambda$, of the input stream $X$. (The full pseudocode of AugmentedMeyerson is available in the supplementary material.) When a new point, $x$, arrives we check whether the $k$-clustering cost of the solution computed on the sketch after adding $x$ to $B_\lambda$ exceeds $\lambda$. If not, we add it to the sketch for $B_\lambda$, if so we reset the $B_\lambda$ substream to $x$, and rename the old sketch of $B_\lambda$ as $A_\lambda$. Thus the algorithm maintains two sketches, on consecutive subintervals. Notice that the cost of each sketch is at most $\lambda$, and each sketch is grown to be maximal before being reset.

We remark that to convert the Meyerson sketch to a $k$-clustering solution, we need to run a $k$-clustering algorithm on the weighted instance given by the sketch. Since the problem is NP-hard, let ALG denote any $\rho$-approximate algorithm, such as the one by [36]. Let $S(Z) = (Y(Z), \mathrm{weight}(Z))$ denote the augmented Meyerson sketch built on a (sub)stream $Z$, with $Y(Z)$ as the centers, and $\mathrm{weight}(Z)$ as the (approximate) weight function. We denote by $\mathrm{ALG}(S(Z))$ the solution obtained by running ALG over the weighted instance $S(Z)$. Let $\widehat{f}_p(S(Z), \mathrm{ALG}(S(Z)))$ be the estimated cost of the solution $\mathrm{ALG}(S(Z))$ over the stream $Z$ obtained by the sketch $S(Z)$.

We show that we can implement a function $\widehat{f}_p$ that operates only on the information in the augmented Meyerson sketch $S(Z)$ and gives a $\beta \in O(\rho)$ approximation to the cost on the unabridged input.

**Lemma 3.2** (Approximate solution and approximate cost from a sketch). *Using an approximation algorithm ALG, from the augmented Meyerson sketch $S(Z)$, with probability $\geq 1 - \gamma$, we can output a solution $\mathrm{ALG}(S(Z))$ and an estimate $\widehat{f}_p(S(Z), \mathrm{ALG}(S(Z)))$ of its cost s.t. $f_p(Z, \mathrm{ALG}(S(Z))) \leq \widehat{f}_p(S(Z), \mathrm{ALG}(S(Z))) \leq \beta(\rho) f_p(Z, \mathrm{OPT}(Z))$ for a constant $\beta(\rho) \leq 2^{3p+6} \rho$ depending only the approximation factor $\rho$ of ALG.*

**Algorithm 1** Meyerson Sketches, $ComputeSketches(X, w, \lambda, m, M, \Delta)$

---

1: Input: A sequence of points $X = x_0, x_1, x_2, \ldots, x_n$. The size of the window $w$. Cost threshold $\lambda$. A lower bound $m$ and upper bound $M$ of the cost of the optimal solution and upper bound on distances $\Delta$.
2: Output: Two sketches for the stream $S_1$ and $S_2$.
3: $S_1 \leftarrow$ AugmentedMeyerson$(\emptyset, w, m, M, \Delta)$; $S_2 \leftarrow$ AugmentedMeyerson$(\emptyset, w, m, M, \Delta)$
4: $A_\lambda \leftarrow \emptyset$; $B_\lambda \leftarrow \emptyset$ (Recall that $A_\lambda, B_\lambda$ are sets and $S_1$ and $S_2$ the corresponding sketches. Note that the content of the sets is not stored explicitly.)
5: **for** $x \in X$ **do**
6:     Let $S_{temp}$ be computed by AugmentedMeyerson$(B_\lambda \cup \{x\}, w, m, M, \Delta)$ . (Note: it can be computed by adding $x$ to a copy of the sketch maintained by $S_2$)
7:     **if** $\widehat{f}_p(S_{temp}, \mathsf{ALG}(S_{temp})) \leq \lambda$ **then**
8:         Add $x$ to the stream of the sketch $S_2$.       ($B_\lambda \leftarrow B_\lambda \cup \{x\}$, $S_2 \leftarrow$ AugmentedMeyerson$(B_\lambda, w, m, M, \Delta)$)
9:     **else**
10:         $S_1 \leftarrow S_2$; $S_2 \leftarrow$ AugmentedMeyerson$(\{x\}, w, m, M, \Delta)$. ($A_\lambda \leftarrow B_\lambda$; $B_\lambda \leftarrow \{x\}$)
11:     **end if**
12: **end for**
13: Return ($S_1, S_2$, and start and end times of $A_\lambda$ and $B_\lambda$)

---

**Composition of sketches from sub-streams**    Before presenting the global sliding window algorithm that uses these pairs of sketches, we introduce some additional notation. Let $S(Z)$ be the augmented Meyerson sketch computed over the stream $Z$. Let $\mathsf{Suffix}_\tau(S(Z))$ denote the sketch obtained from a sketch $S$ for the points that arrived after $\tau$. This can be done using the operations defined in the supplementary material.

We say that a time $\tau$ is contained in a substream $A$ if $A$ contains elements inserted on or after time $\tau$. Finally we define $A_\tau$ as the suffix of $A$ that contains elements starting at time $\tau$. Given two sketches $S(A)$, and $S(B)$ computed over two disjoint substreams $A, B$, let $S(A) \cup S(B)$ be the sketch obtained by joining the centers of $S(A)$ and $S(B)$ (and summing their respective weights) in a single instance. We now prove a key property of the augmented Meyerson sketches we defined before.

**Lemma 3.3** (Composition with a Suffix of stream). *Given two substreams $A,B$ (with possibly $B = \emptyset$) and a time $\tau$ in $A$, let $\mathsf{ALG}$ be a constant approximation algorithm for the $k$-clustering problem. Then if $\mathrm{OPT}_p(A) \leq O(\mathrm{OPT}_p(A_\tau \cup B))$, then, with probability $\geq 1 - O(\gamma)$, we have $f_p(A_\tau \cup B, \mathsf{ALG}(\mathsf{Suffix}_\tau(S(A)) \cup S(B))) \leq O(\mathrm{OPT}_p(A_\tau \cup B))$.*

The main idea of the proof is to show that $\mathsf{Suffix}_\tau(S(A)) \cup S(B)$ is $\epsilon$-consistent with a good mapping from $A_\tau \cup B$ and then by using a technique similar to Lemma 2.1 show that we can compute a constant approximation from an $\epsilon$-consistent sketch.

---

**Algorithm 2** Our main algorithm. Input: $X, m, M, \Delta$, approx. factor of $\mathsf{ALG}$ ($\beta$) and $\delta$.

---

1: $\Lambda \leftarrow \{m, (1+\delta)m, \ldots, 2^p\beta(1+\delta)M\}$
2: **for** $\lambda \in \Lambda$ **do**
3:     $S_{\lambda,1}, S_{\lambda,2} \leftarrow$ ComputeSketches$(X, w, \lambda, m, M, \Delta)$
4: **end for**
5: **if** $B_{\lambda^*} = W$ for some $\lambda^*$ **then return** $\mathsf{ALG}(S_{\lambda^*,2})$
6: $\lambda^* \leftarrow \min(\{\lambda : A_\lambda \not\subseteq W\})$
7: $\tau \leftarrow \max(|X| - w, 1)$
8: **if** $W \cap A_{\lambda^*} \neq \emptyset$ **then return** $\mathsf{ALG}(\mathsf{Suffix}_\tau(S_{\lambda^*,1}) \cup S_{\lambda^*,2})$
9: **else return** $\mathsf{ALG}(\mathsf{Suffix}_\tau(S_{\lambda^*,2}))$

---

**Final algorithm.**    We can now present the full algorithm in Algorithm 2. As mentioned before, we run multiple copies of $ComputeSketches$ in parallel, for geometrically increasing values of $\lambda$.

For each value of $\lambda$, we maintain the pair of sketches over the stream $X$. Finally, we compute the centers using such sketches. If we get lucky, and for the sliding window $W$ there exists a subsequence where $B_{\lambda^*}$ is precisely $W$, we use the appropriate sketch and return $\mathsf{ALG}(S_{\lambda^*,2})$. Otherwise, we find the smallest $\lambda^*$ for which $A_\lambda$ is not a subset of $W$. We then use the pair of sketches associated with $A_{\lambda^*}$ and $B_{\lambda^*}$, combining the sketch of the suffix of $A_{\lambda*}$ that intersects with $W$, and the sketch on $B_{\lambda^*}$.

The main result is that this algorithm provides a constant approximation of the $k$-clustering problem, for any $p \geq 1$, with probability at least $1 - \gamma$, using space linear in $k$ and logarithmic in other parameters. The total running time of the algorithm depends on the complexity of ALG. Let $T(n, k)$ be the complexity of solving an instance of $k$-clustering with size $n$ points using ALG.

**Theorem 3.4.** *With probability $1 - \gamma$, Algorithm 2, outputs an $O(1)$-approximation for the sliding window $k$-clustering problem using space: $O\big(k \log(\Delta)(\log(\Delta) + \log(w) + \log(M))$ $\log^2(M/m) \log(\gamma^{-1} \log(M/m))\big)$ and total update time $O(T(k \log(\Delta), k)$ $\log^2(M/m) \log(\gamma^{-1} \log(M/m)) (\log(\Delta) + \log(w) + \log(M))$.*

We remark that if $M$ and $\Delta$ are polynomial in $w$, then the total space is $O(k \log^4 w \log(\log w/\gamma))$ and the total update time is $O(T(k \log w, k) \log^3(w) \log(\log w/\gamma))$. The main component in the constant approximation factor of Theorem 3.4 statement comes from the $2^{3p+5}\rho$ approximation for the insertion-only case [43]. Here $p$ is the norm, and $\rho$ is the offline algorithm factor. Given the composition operation in our analysis in addition to applying triangle inequality and some other steps, we end up with an approximation factor $\approx 2^{8p+6}\rho$. We do not aim to optimize for this approximation factor, however it could be an interesting future direction.

## 4 Empirical Evaluation

We now describe the methodology of our empirical evaluation before providing our experiments results. We report only the main results in the section, more details on the experiments and results are in supplementary material. Our code is available open-source on github[6]. All datasets used are *publicly-available*.

**Datasets.** We used 3 real-world datasets from the UCI Repository [28] that have been used in previous experiments on $k$-clustering for data streams settings: SKINTYPE [12], $n = 245057, d = 4$, SHUTTLE, $n = 58000, d = 9$, and COVERTYPE [13], $n = 581012, d = 54$. Consistent with previous work, we stream all points in the natural order (as they are stored in the dataset). We also use 4 publicly-available synthetic dataset from [31] (the S-Set series) that have ground-truth clusters. We use 4 datasets (s1, s2, s3, s4) that are increasingly harder to cluster and have each $k = 15$ ground-truth clusters. Consistent with previous work, we stream the points in random order (as they are sorted by ground truth in the dataset). In all datasets, we pre-process each dataset to have zero mean and unit standard deviation in each dimension. All experiments use Euclidean distance, we focus on the the K-MEANS objective ($p = 2$) which we use as cost. We use $k$-means++ [4] as the solver ALG to extract the solution from our sketch.

**Parameters.** We vary the number of centers, $k$, from 4 to 40 and window size, $w$, from 10,000 to 40,000. We experiment with $\delta = [0.1, 0.2]$ and set $\epsilon = 0.05$ (empirically the results are robust to wide settings of $\epsilon$).

**Metrics.** We focus on three key metrics: cost of the clustering, maximum space requirement of our sketch, and average running time of the update function. To give an implementation independent view into space and update time, we report as space usage the number of points stored, and as update time the number of distance evaluations. All of the other costs are negligible by comparison.

**Baselines.** We consider the following baselines.
**Batch K-Means++**: We use $k$-means++ over the entire window as a proxy for the optimum, since the latter is NP-hard to compute. At every insertion, we report the best solution over 10 runs of $k$-means++ on the window. Observe that this is inefficient as it requires $\Omega(w)$ space and $\Omega(kw)$ run time per update. **Sampling**: We maintain a random sample of points from the active window, and then run $k$-means++ on the sample. This allows us to evaluate the performance of a baseline, at the same space cost of our algorithm. **SODA16**: We also evaluated the only previously published algorithm for this setting in [17].

We note that we made some practical modifications to further improve the performance of our algorithm which we report in the supplementary material.

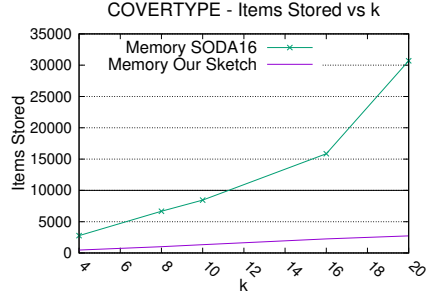

Figure 1: Comparison of the max memory use of SODA16 and our algorithm for $W = 10,000$.

| Dataset | $k$ | Space Decr. Factor | Speed-Up Factor | Cost (ratio) |
|---|---|---|---|---|
| COVER | 4 | 5.23 | 10.88 | 99.5% |
| | 16 | 6.04 | 13.10 | 95.1% |
| SHUTTLE | 4 | 5.07 | 9.09 | 106.8% |
| | 16 | 15.32 | 15.14 | 118.9% |

Table 1: Decrease of space use, decrease in (speed-up) and ratio of mean cost of the solutions of our algorithm vs the SODA16 baseline (100% means same cost, $< 100\%$ means a reduction in cost).

**Comparison with previous work.** We begin by comparing our algorithm to the previously published algorithm of [17]. The baseline in this paragraph is **SODA16** algorithm in [17]. We confirm empirically that the memory use of this baseline already exceeds the size of the sliding window for very small $k$, and that it is significantly slower than our algorithm. Figure 1 shows the space used by our algorithm and by the baseline over the COVERTYPE dataset for a $|W| = 10,000$ and different $k$. We confirm that our algorithm's memory grows linearly in $k$ while the baseline grows super-linearly in $k$ and that for $k > 10$ the baseline costs more than storing the entire window. In Table 1 we show that our algorithm is significantly faster and uses less memory than the **SODA16** already for small values of $k$. In the supplementary material we show that the difference is even larger for bigger values of $k$. Given the inefficiency of the SODA16 baseline, for the rest of the section we do not run experiments with it.

**Cost of the solution.** We now take a look at how the cost of the solution evolves over time during the execution of our algorithm. In Figure 2 we plot the cost of the solution obtained by our algorithm (Sketch), our proxy for the optimum (KM++) and the sampling baseline (Sampling Baseline) on the COVERTYPE dataset. The sampling baseline is allowed to store the same number of points stored by our algorithm (at the same point in time). We use $k = 20$, $|W| = 40,000$, and $\delta = 0.2$. The plot is obtained by computing the cost of the algorithms every 100 timesteps. Observe that our algorithm closely tracks that of the offline algorithm result, even as the cost fluctuates up and down. Our algorithm's cost is always close to that of the off-line algorithm and significantly better than the random sampling baseline

**Update time and space tradeoff.** We now investigate the time and space tradeoff of our algorithm. As a baseline we look at the cost required simply to recompute the solution using $k$-means++ at every time step. In Table 2 ($\delta = 0.2$) we focus on the COVERTYPE dataset, the other results are similar. Table 2 shows the percent of the sliding window data points stored (Space) and the percent of update

| $W$ | $k$ | Space | Time |
|---|---|---|---|
| | 10 | 3.5% (0.39%) | 0.45% (0.29%) |
| 40000 | 20 | 6.5% (0.87%) | 0.93% (0.63%) |
| | 40 | 11.3% (1.74%) | 1.58% (1.23%) |

Table 2: Max percentage of sliding window (length $W$) stored (Space) and median percentage of time (Time) vs. one run of $k$-means++ (stddev in parantesis).

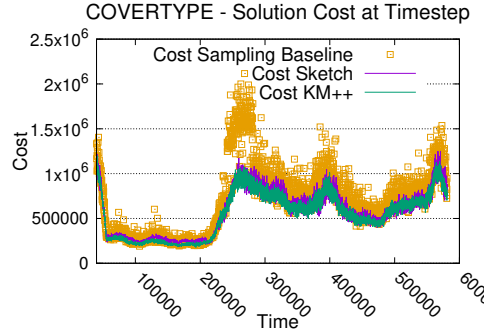

Figure 2: Cost of the solution obtained by our algorithm (Sketch) and the two baselines for $k = 20$, $|W| = 40{,}000$ and $\delta = 0.2$ on COVERTYPE.

time (Time) of our algorithm vs a single run of $k$-means++ over the window. In the supplementary material we show that the savings become larger (at parity of $k$) as $|W|$ grows and that we always store a small fraction of the window, providing order-of-magnitude speed ups (e.g., we use $< 0.5\%$ of the time of the baseline for $k = 10$, $|W| = 40{,}000$). Here the baseline is the $k$-means++ algorithm.

**Recovering ground-truth clusters.** We evaluated the accuracy of the clusters produced by our algorithm on a dataset with ground-truth clusters using the well known V-Measure accuracy definition for clustering [51]. We observe that on all datasets our algorithm performs better than the sampling baseline and in line with the offline $k$-means++. For example, on the s1 our algorithm gets V-Measure of $0.969$, while $k$-means++ gets $0.969$ and sampling gets $0.933$. The full results are available in the supplementary material.

# 5    Conclusion

We present the first algorithms for the $k$-clustering problem on sliding windows with space linear in $k$. Empirically we observe that the algorithm performs much better than the analytic bounds, and it allows to store only a small fraction of the input. A natural avenue for future work is to give a tighter analysis, and reduce this gap between theory and practice.

## Broader Impact

Clustering is a fundamental unsupervised machine learning problem that lies at the core of multiple real-world applications. In this paper, we address the problem of clustering in a sliding window setting. As we argued in the introduction, the sliding window model allows us to discard old data which is a core principle in data retention policies.

Whenever a clustering algorithm is used on user data it is important to consider the impact it may have on the users. In this work we focus on the algorithmic aspects of the problem and we do not address other considerations of using clustering that may be needed in practical settings. For instance, there is a burgeoning literature on fairness considerations in unsupervised methods, including clustering, which further delves into these issues. We refer to this literature [22, 40, 11] for addressing such issues.

## Funding Transparency Statement

No third-party funding has been used for this research.

## Footnotes

[1]We note that the authors assume that the cost of any solution is polynomial in $w$. We chose to state our bounds explicitly, which introduces a dependence on the ratio of the max and min costs of the solution.

[2]We note here that in some practical applications $k$ can be large. For instance, in spam and abuse [53], near-duplicate detection [37] or reconciliation tasks [52].

[3]In the Euclidean space, if the centers do not need to be part of the input, then setting $p = 2$ recovers the $k$-MEANS problem.

[4]These assumptions are not necessary. In the supplementary material, we explain how we estimate them in our experiments and how from a theoretical perspective we can remove the assumptions.

[5] We note that prior work [17, 25] makes similar assumptions to get a bound depending on $w$.

[6]https://github.com/google-research/google-research/tree/master/sliding_window_clustering/

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
