[Supplementary Material]

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

[7]We do not optimize constants throughout the paper to keep proofs simpler. We empirically evaluate the algorithm's performance and show the efficacy of our approach.

[8] Note that we can do this because no point has weight larger than $W$ inside the sliding window at any point in the algorithm.

[9]Note that we can do this because mapping of a good Meyerson sketch cost more than $(1 + \epsilon)2^{p+7}M$ by Lemma B.1.

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

## A Omitted Proofs from Preliminaries

**Lemma A.1** (Restatement of Lemma 2.1). *Given a set of points $X = \{x_1, \ldots, x_n\}$ consider a multiset $Y = \{y_1, \ldots, y_n\}$ such that $\sum_i d^p(x_i, y_i) \leq \alpha \operatorname{OPT}_p(X)$, for a constant $\alpha$. Let $\mathcal{B}^*$ be the optimal solution for $Y$. Then $f_p(X, \mathcal{B}^*) \in O(OPT_p(X))$.*

*Proof.* This proof follows through several applications of triangle inequality. First, observe that:

$$f_p(X, \mathcal{B}^*) = \sum_i d^p(x_i, \mathcal{B}^*) \leq 2^{p-1} \sum_i \Big( d^p(x_i, y_i) + d^p(y_i, \mathcal{B}^*) \Big) \tag{1}$$

instance $(X, d)$ as $\operatorname{OPT}_p(X)$, and the optimal clustering as $\mathcal{C}_p^*(X) = \{c_1^*, c_2^*, \ldots, c_k^*\}$ , shortening to $\mathcal{C}^*$ when clear from context. Notice that everywhere in the paper, we assume that $p$ is a constant with $p \geq 1$.

Let $\mathcal{C}^* = \{c_1^*, \ldots, c_k^*\}$ be the optimal solution for $X$. In a slight abuse of notation, let $\mathcal{C}^*(x_i)$ represent the optimal center closest to $x_i$. In addition, for every $c_j^*$, let $t_j$ be its closest neighbor in $Y$, $t_j = \operatorname{cl}(c_j^*, Y)$. Observe that the set $T = \{t_1, \ldots, t_k\}$ represents a feasible solution.

Therefore:

$$\sum_i d^p(y_i, B^*) \leq d^p(y_i, T). \tag{2}$$

Finally, let $t(i) = \operatorname{cl}(\mathcal{C}^*(x_i), Y)$ represent the point in $Y$ which is closest to the optimal center assigned to $x_i$.

$$\begin{aligned}
\sum_i d^p(y_i, \mathcal{B}^*) &\leq \sum_i d^p(y_i, t(i)) \\
&\leq 2^{p-1} \sum_i \Big( d^p(y_i, \mathcal{C}^*(x_i)) + d^p(\mathcal{C}^*(x_i), t(i)) \Big) \\
&\leq 2^{p-1} \sum_i 2 \cdot d^p(y_i, \mathcal{C}^*(x_i)) \\
&\leq 2^{2p-1} \Big( \sum_i d^p(y_i, x_i) + d^p(x_i, \mathcal{C}^*(x_i)) \Big) \\
&\leq 2^{2p-1}(1 + \alpha)OPT_p(X),
\end{aligned}$$

Where the first inequality follows because $T$ is a feasible solution, the second and fourth by the generalized triangle inequality, and the third by the definition of $t(i)$. Putting this together with Equation 1 completes the proof. □

For completeness we present the next lemma that would be useful to show our main theorem

**Lemma A.2.** *Given a set of points $X = \{x_1, \ldots, x_n\}$ by adding points to the set $X$ the optimal cost of the $k$-clustering can decrease at most by a $2^p$.*

*Proof.* Let $Y = \{x_1, \ldots, x_n, y_1, \ldots, y_m\}$ be the set obtained by adding $\{y_1, \ldots, y_m\}$ arbitrary points to X. Let $\mathcal{C}^* = \{c_1^*, c_2^*, \ldots, c_k^*\}$ be the optimal set of centers for $Y$ and let $\mathcal{C}' = \{x_{c_1^*}, \ldots, x_{c_k^*}\}$ the (multi-)set where $x_{c_i^*}$ is the closest point to $c_i^*$ in $X$. Furthermore let $c^*(x)$ be the closest center in $\mathcal{C}^*$ to a point $x$. Then we have $\operatorname{OPT}_p(X) \leq \sum_{x \in X} d(x, \mathcal{C}') \leq \sum_{x \in X} 2^{p-1}(d(x, c^*(x)) + d(c^*(x), x_{c^*(x)}) \leq \sum_{x \in X} 2^p d(x, c^*(x)) \leq 2^p \sum_{x \in X} d(x, \mathcal{C}^*) \leq 2^p \sum_{y \in Y} d(y, \mathcal{C}^*) \leq 2^p \operatorname{OPT}_p(Y)$. □

## B Meyerson Sketches

In this section we define the Meyerson sketch algorithm and show some basic facts about the property of any mapping $\mu$ and properties of the Meyerson sketch.

The results presented here follow from well-known results [20, 44], but we present them here for completeness.

### B.1 Meyerson algorithm

We begin by formally stating the guarantees of the Meyerson sketch, giving a constant factor approximation for any constant $p$.[7] The following Lemma, captures the main properties of the sketch.

**Lemma B.1.** *Let $(X, \mathrm{d})$ be a metric space and fix $\gamma \in (0, 1)$. Then the Meyerson sketch algorithm computes a mapping $\mu : X \to \mathcal{C}$, and a consistent weighted instance $(\mathcal{C},\text{weight})$, such that $\mathcal{C} \subseteq X$ and, with probability $1 - \gamma$: $|\mathcal{C}| \leq 2^{2p+8}k \log {}^1\!/_\gamma \log \Delta$; $f_p(X, \mathcal{C}) \leq 2^{p+7} \operatorname{OPT}_p(X)$. The algorithm uses at most $2^{2p+8}k \log {}^1\!/_\gamma \log \Delta \lceil \log M/m \rceil$ space and in the consistent mapping $\mu$, each point is mapped to a center inserted before it into the stream.*

We now present the pseudocode of the Meyerson sketch algorithm. Fix the parameter $p$ of the problem. Consider for now to have access to a lower bound to the cost $L^p$ of the optimal solution for the problem, such that $L^p \geq \alpha \operatorname{OPT}_p$, for some constant $0 < \alpha < 1$. We will remove this assumption at the end of the section.

We start by presenting the code for a single Meyerson sketch as described in Algorithm 3. It picks each point as a new center with the probability proportional to the distance of the point from the selected centers so far. We note that a point that is far from the current centers should be be added to avoid a large cost for that point. To obtain the results in Lemma B.1 we will run multiple instances of Algorithm 3.

---

**Algorithm 3** Single Meyerson sketch

---

1: Input: A sequence of points $x_0, x_1, x_2, \ldots, x_n$. A finite $p$, a guess $L^p$, an upper bound $\Delta$ of the max distance, and the parameter $k$.
2: Output: A mapping $\mu : X \to S$, and a weighted instance $(S, \text{weight})$ that is consistent with $X$ and cost of the moving $\text{cost}_\mu$.
3: $S \leftarrow \emptyset$
4: Let $X$ be a set of points and assume $L^p$ is such that $L^p \geq \alpha \operatorname{OPT}_p(X)$, for some constant $0 < \alpha < 1$
5: $\text{cost}_\mu = 0$
6: **for** $x \in X$ **do**
7:     **if** $S = \emptyset$ **then**
8:         $S \leftarrow \{x\}$
9:     **else**
10:         With probability $\min\left(\frac{k(1+\log \Delta)d(x,S)^p}{L^p}, 1\right)$ add $x$ to $S$ and set $\text{weight}(x) = 1$
11:         Otherwise, let $z \leftarrow \operatorname{argmin}_{y \in S} d(x, y)$, set $\text{weight}(z) \leftarrow \text{weight}(z) + 1, \text{cost}_\mu \leftarrow \text{cost}_\mu + d(x, y)^p$
12:     **end if**
13: **end for**
14: Return $S$, weight, $\text{cost}_\mu$

---

Using a single sketch we can show the following property.

**Lemma B.2.** *For a constant $\alpha \in (0, 1)$ and input $X$, Algorithm 3 computes a mapping $\mu$ and a consistent weighted instance $(S, \text{weight})$ with probability at least $\frac{1}{2}$:*

$$|S| \leq 4k(1 + \log \Delta)\left(\frac{2^{p+3}}{\alpha^p} + 1\right) \quad and$$

$$f_p(X, S) \leq \text{cost}_\mu \leq 2^{p+5} \operatorname{OPT}_p(X).$$

*Furthermore, in the consistent mapping $\mu$ between $X$ and $S$, every point in $X$ is mapped to a point in $S$ inserted before it into the stream. The algorithm computes the cost of the mapping $\text{cost}_\mu$.*

*Proof.* Our mapping $\mu$ is defined by $z = \operatorname{argmin}_{y \in S} d(x, y)$ in Algorithm 3, note that every point is mapped to a point inserted earlier than it. We will show that in expectation $S$ has size $k(1 + \log \Delta)\left(\frac{2^{2p+1}}{\alpha^p} + 1\right)$ and $f_p(X, S) \leq 2^{p+3} \operatorname{OPT}_p(X)$. The lemma then follows by an application of Markov inequality. Let $c_1^*, c_2^*, \ldots, c_k^*$ be the centers in the optimal solution, $C_1^*, C_2^*, \ldots, C_k^*$ be the respective clusters and $\mathcal{C}^*$ be the optimal clustering. Let $f_p(C_i^*, \mathcal{C}^*)$ be the cost for cluster $i$ and $a_i^*$ be: $a_i^* = \left(\frac{f_p(C_i^*, \mathcal{C}^*)}{|C_i^*|}\right)^{1/p}$.

Consider the set of points in $C_i^*$ with distance at most $a_i^*$ to the center $\mathsf{c}_i^*$. Let $D = \{v_1, v_2, \ldots v_q\}$ be the set of points with distance at most $a_i^*$ to the center $\mathsf{c}_i^*$. Let $p_i$ be the probability that element $v_i$ is added to $S$, i.e., $p_i = \min(1, \frac{d(v_i,S)^p k(1+\log \Delta)}{L^p})$. With some abuse of notation we say that $p_{q+1} = 1$. We want to estimate the cost of the subset of $D$ in the stream before any point in $D$ is added to $S$. For the points $v_1, \ldots, v_{q'}$, $q' \leq q$, before the first point added to the set, the probability of them being added to the set is $\frac{d(v_i,S)^p k(1+\log \Delta)}{L^p} < 1$ and its cost is $d(v_i, S)^p$. So the expected cost for these points is:

$$
\begin{aligned}
&\sum_{i=1}^{q'} (\sum_{j=1}^{i} d(v_j, S)^p) \left( p_{i+1} \prod_{j=1}^{i} (1 - p_j) \right) \\
&= \sum_{i=1}^{q'} d(v_i, S)^p \left( \prod_{j=1}^{i} (1 - p_j) \right) \left( \sum_{j=i+1}^{q'+1} p_j \prod_{l=i+1}^{j-1} (1 - p_l) \right) \\
&\leq \sum_{i=1}^{q'} d(v_i, S)^p \left( \prod_{j=1}^{i} (1 - p_j) \right) \\
&\leq \frac{L^p}{k(1 + \log \Delta)} \sum_{i=1}^{q'} \prod_{j=1}^{i} (1 - p_j) \leq \frac{L^p}{k(1 + \log \Delta)}.
\end{aligned}
$$

The first inequality follows from the fact that we are dividing by a number smaller than $1$ ($\frac{d(v,S)^p k(1+\log \Delta)}{L^p} < 1$).

After we add a point to the set, we can use the generalized triangle inequality to bound the cost of any point $v$ in the set with $\left( 2^{p-1}((a_i^*)^p + d(v, \mathsf{c}_i^*)^p) \right)$ Therefore, the probability of adding $v$ as a center is bounded by $\left( 2^{p-1}((a_i^*)^p + d(v, \mathsf{c}_i^*)^p) \right) k(1 + \log \Delta)/L^p$.

Similarly for any $j > 0$ when we consider the points in $C_i^*$ that have distance between $2^j a_i^*$ and $2^{j+1} a_i^*$ to the center $\mathsf{c}_i^*$, the expected cost for all points added before inserting any element to $S$ is at most $\left( \frac{L^p}{k(1+\log \Delta)} \right)$. After we add a point from this annulus to set $S$, we can use the generalized triangle inequality to bound the cost of any other point $v$ by $\left( 2^{p-1}((2^{j+1} a_i^* + d(v, \mathsf{c}_i^*)^p)) \right) \leq \left( 2^{p-1}((2d(v, \mathsf{c}_i^*))^p + d(v, \mathsf{c}_i^*)^p) \right)$. Again, the probability of adding this point as a center is $\frac{\left( 2^{p-1}(2d(v,\mathsf{c}_i^*))^p + d(v,\mathsf{c}_i^*)^p \right) k(1+\log \Delta)}{L^p}$.

Therefore, we can bound the expected number of centers added by:

$$
\begin{aligned}
&\sum_{C_i^*} \left( 1 + \log \Delta + \frac{k(1 + \log \Delta)}{L^p} \sum_{v \in C_i^*} 2^{p-1} \left( (a_i^*)^p + (2d(v, \mathsf{c}_i^*))^p + d(v, \mathsf{c}_i^*)^p \right) \right) \\
&\leq \sum_{C_i^*} \left( 1 + \log \Delta + \frac{k(1 + \log \Delta)}{L^p} 2^{p+1} \sum_{v \in C_i^*} \left( (a_i^*)^p + d(v, \mathsf{c}_i^*)^p \right) \right) \\
&\leq k(1 + \log \Delta) + \sum_{C_i^*} \left( k(1 + \log \Delta) 2^{p+2} \frac{f_p(C_i^*, \mathcal{C}^p)}{L^p} \right) \\
&= k(1 + \log \Delta) + \left( k(1 + \log \Delta) 2^{p+2} \frac{\mathrm{OPT}_p(X)}{L^p} \right) \\
&\leq \left( 1 + \frac{2^{p+3}}{\alpha^p} \right) k(1 + \log \Delta).
\end{aligned}
$$

In a similar manner, we can bound the cost by

$$\sum_{C_i^*} \left( \frac{L^p}{k(1+\log\Delta)}(1+\log\Delta) + \sum_{v\in C_i^*} 2^{p-1}\Big( (a_i^*)^p + (2d(v,\mathsf{c}_i^*))^p + d(v,\mathsf{c}_i^*)^p \Big) \right)$$

$$\leq \sum_{C_i^*} \left( \frac{L^p}{k} + 2^{p+1} \sum_{v\in C_i} ((a_i^*)^p + d(v,\mathsf{c}_i^*)^p) \right)$$

$$\leq (L^p + 2^{p+2}\,\mathrm{OPT}_p(X)) \leq 2^{p+3}\,\mathrm{OPT}_p(X)$$

Thus the claim follows. $\qquad\square$

First step in proving Lemma B.1 using Lemma B.2 is to show how to get our guarantees with probability $1-\gamma$. This is done in Algorithm 4. Lemma B.3 states the properties of this algorithm.

---

**Algorithm 4** $ComputeMeyerson(X, L^p, \alpha, \gamma, \Delta, p, k)$

1: **Input:** A sequence of points $X$, a lower bound to the optimum $L^p$, a constant $\alpha$ such a that $L_p \geq \alpha\,\mathrm{OPT}_p$, a constant $\gamma$, an upper bound $\Delta$ on the max distance, the parameters $p$ and $k$ of the problem.
2: **Output:** A mapping $\mu : X \to M$, and a weighted instance $(M, \mathsf{weight})$ that is consistent with $X$ and cost of the moving $\mathsf{cost}_\mu$
3: **for** $i \in [2\log\gamma^{-1}]$ **do**
4: $\quad M_i \leftarrow x_0$
5: $\quad \mathsf{cost}_{\mu_i} = 0$
6: **end for**
7: **for** $x \in X_t$ **do**
8: $\quad$ **for** $i \in [2\log\gamma^{-1}]$ **do**
9: $\qquad$ **if** $|M_i| \leq 4k(1+\log\Delta)\left(\frac{2^{p+3}}{\alpha^p}+1\right)$ **then**
10: $\qquad\quad$ **if** $M_i == \emptyset$ **then**
11: $\qquad\qquad M_i \leftarrow \{x\}$
12: $\qquad\quad$ **else**
13: $\qquad\qquad$ With probability $\min\left( \frac{k(1+\log\Delta)d(x,S)^p}{L^p}, 1 \right)$ add $x$ to $M_i$ and set $\mathsf{weight}_i(x) = 1$
14: $\qquad\qquad$ Otherwise, let $z \leftarrow \mathrm{argmin}_{y\in S} d(x,y)$, set $\mathsf{weight}_i(z) \leftarrow \mathsf{weight}_i(z)+1$, $\mathsf{cost}_{\mu_i} \leftarrow \mathsf{cost}_{\mu_i} + d(x,y)^p$
15: $\qquad\quad$ **end if**
16: $\qquad$ **end if**
17: $\quad$ **end for**
18: **end for**
19: Let $j$ be the index of the Meyerson sketch of minimum cost $\mathsf{cost}_{\mu_j}$ such that $|M_j| \leq 4k(1+\log\Delta)\left(\frac{2^{p+3}}{\alpha^p}+1\right)$, if such $j$ does not exist return $M = \cup_{i=1}^{2\log\gamma^{-1}} M_i$, $\mathsf{weight}_1, \infty$
20: Extend $\mathsf{weight}_j$ to give weight 0 to all the points in $M = \cup_{i=1}^{2\log\gamma^{-1}} M_i$ not contained in $M_j$
21: Return $M = \cup_{i=1}^{2\log\gamma^{-1}} M_i$, $\mathsf{weight}_j$, $\mathsf{cost}_{\mu_j}$

---

**Lemma B.3.** *For a constant $\alpha \in (0,1)$ and input $X$, Algorithm 4 computes a mapping $\mu$ and a consistent weighted instance $M = \cup_{i=1}^{2\log\gamma^{-1}}(M_i, \mathsf{weight}_i)$ such that with probability at least $1-\gamma$, we have:*

$$|M| \leq 8k\log\gamma^{-1}(1+\log\Delta)\left( \frac{2^{p+3}}{\alpha^p}+1 \right) \in O(k\log\Delta\log\gamma^{-1})$$

$$and \qquad f_p(X,M) \leq 2^{p+5}\,\mathrm{OPT}_p(X)$$

*Furthermore, in the consistent mapping $\mu$ between $X$ and $M$ every point in $X$ is mapped to a point in $M$ inserted before it into the stream. The algorithm computes the cost of the mapping $\mathsf{cost}_\mu$.*

*Proof.* As mentioned above, Lemma B.2 implies that if we construct $2\log\gamma^{-1}$ single Meyerson sketches in parallel, with probability in $1-\gamma$, at least one of them gives a constant approximation to the optimum at every point in time and furthermore the single Meyerson contains only $4k(1+\log\Delta)\left( \frac{2^{p+3}}{\alpha^p}+1 \right)$ points.

Now in Algorithm 4 we are almost building $2\log\gamma^{-1}$ single Meyerson sketches, the only difference is that we stop adding points to a single sketch if that becomes too large. This modification does not change the probability that there exist at least one single sketch that gives a constant approximation to the optimum at every point in time and that contains only $4k(1+\log\Delta)\left(\frac{2^{p+3}}{\alpha^p}+1\right)$ points.

Thus with probability $1-\gamma$ at least one of the sketches constructed in B.2 gives a constant approximation to the optimum at every point in time. Merging other sketches to this sketch does not affect this property. Furthermore the number of points in each sketch is explicitly bounded by $4k(1+\log\Delta)\left(\frac{2^{p+3}}{\alpha^p}+1\right)$ so the total number of points in $M$ is bounded by $8k\log\gamma^{-1}(1+\log\Delta)\left(\frac{2^{p+3}}{\alpha^p}+1\right)$.

Now we have to prove the existence of a consistent mapping, note that for this we can just use the weighting of a Meyerson sketch with less $4k(1+\log\Delta)\left(\frac{2^{p+3}}{\alpha^p}+1\right)$ and cost less $2^{p+5}\,\mathrm{OPT}_p(X)$ and assign the weight to all the other points equal to 0. $\qquad\square$

Finally in order to complete the proof of Lemma B.1, we need to remove the assumption of knowing a good lower bound for the optimal solution. We do so by trying different guess of the $OPT$ in particular we try for different $L^p$ in $\{m, 2m, 4m, 8m, \ldots, 2^{\lceil\log M/m\rceil}m\}$. This guarantees that we also run it for a guess $L^p$ such that $L^p \geq \frac{1}{2}\,\mathrm{OPT}_p(X)$. Now we can detect the correct guess by checking the cost of the Meyerson sketches. The pseudo-code of the final algorithm is presented in Algorithm 5.

---

**Algorithm 5** $SimpleMeyerson(X, m, M, \gamma, \Delta, p, k)$

1: **Input:** A sequence of points $X$, lower bound $m$ and upper bound $M$ to the optimum, and $\gamma$, an upper bound $\Delta$ on the max distance, the parameters $p$ and $k$ of the problem.
2: **Output:** A mapping $\mu : X \to X'$, and a weighted instance $(X', \mathsf{weight})$ that is consistent with $X$ with cost of the moving $\mathsf{cost}_\mu$
3: **for** $L^p \in \{m, 2m, 4m, 8m, \ldots, 2^{\lceil\log M/m\rceil}m\}$ **do**
4: $\quad$ In parallel $ComputeMeyerson(X, L^p, \alpha = 1/2, \gamma, \Delta, p, k)$
5: **end for**
6: Let $\ell$ be the smallest index (if it exists, otherwise it is an arbitrary index) in $\{m, 2m, 4m, 8m, \ldots, 2^{\lceil\log M/m\rceil}m\}$ for which the output $ComputeMeyerson$ called with $L_p = \ell$ has size smaller than $8k\log\gamma^{-1}(1+\log\Delta)\left(2^{2p+3}+1\right)$ and cost smaller than $2^{p+6}\ell$
7: Return the result of the index $\ell$ call for $ComputeMeyerson$.

---

We are now ready to prove the main Lemma of this section.

*Proof of Lemma B.1.* The bound on the size of the set and cost follow from the check in Algorithm 5. The probability of success is larger than or equal to the probability of success for the $L^p \in \{m, 2m, 4m, 8m, \ldots, 2^{\lceil\log M/m\rceil}m\}$ such that $L^p \geq \frac{1}{2}\,\mathrm{OPT}_p(X)$. The total space is a result of running $O(\log M/m)$ times $ComputeMeyerson$ calls in parallel and the result in Lemma B.3. $\qquad\square$

## C  Augmented Meyerson Sketches

We now show how we can augment the sketch data structure with additional information for clustering in the sliding window model.

### C.1  Maintaining Weights.

As we mentioned above, one of the challenges in adapting the Meyerson sketch to the sliding window setting lies in tracking the weight of a center as points expire. In fact, some of the points mapped initially to a center by $\mu$ may no longer be part of the sliding window. Formally, given a stream $X = x_1, x_2, \ldots, x_t$, we would like to maintain an estimate of the weight of each center in $\mathcal{C}$ restricted to the sliding window, i.e., for $\mathsf{c} \in \mathcal{C}$, we want to estimate $|\{x \in W : \mu(x) = \mathsf{c}\}|$. We denote this

quantity by $\mathsf{weight}_\mu(\mathsf{c}, X_{[t-w,t]})$. Maintaining an estimate of such a weight function falls directly into the Smooth Histograms framework introduced by [18]. We leverage their approach to maintain a $(1 + \epsilon)$ estimate using only $\log_{1+\epsilon} w$ additional overhead per cluster center.

**Lemma C.1.** *Fix constant $\epsilon > 0$. Using additional space $O\left(|\mathcal{C}| \log_{1+\epsilon}(w)\right)$, we can extend the Meyerson sketch $\mathcal{C}$, weight with a function $\widehat{\mathsf{weight}} : \mathcal{C} \times [n] \to \mathbb{Z}$ such that, at every time $t$, for every $\mathsf{c} \in \mathcal{C}$, and every time $\tau$ in the active window:* $\mathsf{weight}_\mu(\mathsf{c}, X_{[\tau,t]}) \leq \widehat{\mathsf{weight}}(\mathsf{c}, \tau) \leq (1 + \epsilon)\, \mathsf{weight}_\mu(\mathsf{c}, X_{[\tau,t]})$.

*Proof.* In the sketch, for each point $v \in C$, we maintain a sequence of weights $R_v = (r_{v,1}, r_{v,2}, \ldots)$ corresponding to the number of points assigned to $v$ by $\mu$ and a sequence of times $T_v = (t_{v,1}, t_{v,2}, \ldots)$. The two sequences are initialized as empty. We preserve the invariant that for each time $t_{v,j} \in T_v$, the number of points assigned to center $v$ by $\mu$ from time $t_{v,j}$ (inclusive) to the end of the stream is equal to $r_{v,j}$. To do so, when at time $i$, a point is assigned to $v$ by $\mu$ (recall that the assignment is fixed), we increase by one all of the weights stored in $R_v$ and then we add a new weight initialized to 1 to $R_v$ and a new time equal to $i$ to $T_v$.

To reduce the size of the structure, we maintain only the significant changes in weights in $R_v$. More precisely, at any time, we delete $r_{v,l}$ and $t_{v,l}$ for any $l \in [2, \ldots, |R_v| - 1]$, if $r_{v,l-1} \leq (1+\epsilon)r_{v,l+1}$. We also renumber the indices to be consecutive. Finally, we remove the $r_{v,l}$ (and the corresponding $t_{r,l}$) for which the weights are larger than $(1+\epsilon)|W|$.[8] Notice that at any time, for each $v$ and $l$, either $r_{v,l} = r_{v,l+1} + 1$ or $r_{v,l} \leq (1+\epsilon)r_{v,l+1}$. In fact, if $r_{v,l}$ and $r_{v,l+1}$ refer to consecutive assignments of points to the center, the first case is true. If $r_{v,l}$ and $r_{v,l+1}$ became consecutive after the removal of a point between them, the latter condition is true at the time of the removal, and is preserved by adding 1 to both elements.

Now, in order to answer $\widehat{\mathsf{weight}}(\mathsf{c}, \tau)$ we return the value $r_{i_v}$ in the corresponding $R_v$ array, where $i_v$ is the index of the largest value smaller or equal to $\tau$ in $T_v$. Then $\mathsf{weight}_\mu(\mathsf{c}, X_{[\tau,t]}) \leq \widehat{\mathsf{weight}}(\mathsf{c}, \tau) \leq (1 + \epsilon)\, \mathsf{weight}_\mu(\mathsf{c}, X_{[\tau,t]})$. Finally, note that for each $l \in [|R_v| - 2]$, $r_{v,l} > (1+\epsilon)r_{v,l+2}$, hence, the sequence is decreasing by a factor of $(1 + \epsilon)$ every 2 steps, so that the total length is at most $O(\log_{1+\epsilon}(w))$. □

## C.2 Maintaining Centers.

Our second task is making sure each cluster has a good center. Once again, this issue is unique to the sliding window setting, as in traditional data streams, points (and thus centers) never expire. Specifically, whenever a center $\mathsf{c}$ expires, we aim to replace it with a center $\mathsf{c}'$ such that, $\mathrm{d}(\mathsf{c}, \mathsf{c}')$ is small. In our context, small means comparable to the distance between the center $\mathsf{c}$ and any point $x$ in the window that was mapped to it.

**Definition C.2.** *Fix a mapping $\mu$ and a center $\mathsf{c}$. We say that $y \in X_{[\tau,t]}$ is an $\epsilon$-replacement for $\mathsf{c}$ at time $\tau$ if: $\mathrm{d}(\mathsf{c}, y)^p \leq (1 + \epsilon)\, \mathrm{d}(\mathsf{c}, x)^p$ for any $x \in X_{[\tau,t]}$ with $\mu(x) = \mathsf{c}$, where $t$ is the last time in the current stream.*

To obtain an $\epsilon$-replacement, we consider a sequence of concentric shells with geometrically increasing radii around each center $\mathsf{c}$, and keep the last occurring point in every shell. Since distances are between 1 and $\Delta$, we can bound the total number of shells by $O(\log_{1+\epsilon} \Delta)$ per center. When $\mathsf{c}$ needs to be replaced, we simply look for the smallest non-empty shell and return its representative. By construction, we will ensure that the $\epsilon$-replacement property is satisfied.

**Lemma C.3.** *Let $\epsilon \in (0, 1)$, then with space $O(|\mathcal{C}| \log_{1+\epsilon}(\Delta))$, it is possible to construct a function $\mathsf{last} : \mathcal{C} \times [n] \to X$ that will always return an $\epsilon$-replacement for $\mathsf{c} \in \mathcal{C}$.*

*Proof.* Recall that the minimum distance between two distinct points is 1, and that $\Delta$ is the maximum distance between two points in the dataset. Let $\Lambda$ to be the set $\{1, (1+\epsilon), (1+\epsilon)^2, \ldots \Delta^p\}$ with both extremes included. For each point $y \in C$, we maintain a sequence of elements $L_y = \{l_{y,\lambda} | \lambda \in \Lambda\}$ and a sequence of times $H_y = \{h_{y,\lambda} | \lambda \in \Lambda\}$ indexed by $\lambda \in \Lambda$. When a point $y$ is added to $\mathcal{C}$, we

initialize the sequence $L_y$ with $y$ and $H_y$ with its insertion time. Whenever a point $x$ is mapped by the Meyerson sketch to $y$ at time $i$, let $d = \mathrm{d}(y, x)^p$, we set $l_{y,\lambda} = x$ and $h_{y,\lambda} = i$ for all $\lambda \geq d$. First note that we never add a point $x$ with distance larger than $\Delta$ by definition, so each point is added to at least one set.

Now, to answer $\mathsf{last}(y, \tau)$, we return the point $l_{y,\lambda}$ for the smallest $\lambda$ such that $h_{y,\lambda} \geq \tau$. If no such $\lambda$ exists, we return $\emptyset$.

Note that by construction, we return a point in $\{x_\tau, \ldots, x_t\}$ that is mapped to $y$ if and only if the set is non-empty. Now, suppose that we return $y'$ and $y' \in l_{y,\lambda}$. We know that there exists no point $x \in \cap \{x_\tau, \ldots, x_t\}$ mapped to $y$ by the Meyerson sketch such that $\mathrm{d}(x, y)^p < \frac{\lambda}{1+\epsilon}$; otherwise, the point would be stored in $l_{y, \frac{\lambda}{1+\epsilon}}$, which is a contradiction as we returned $l_{y,\lambda}$. Hence, we have $\mathrm{d}(y, y')^p \leq (1 + \epsilon) \mathrm{d}(x, y)^p$ for any $x \in \cap \{x_\tau, \ldots, x_t\}$ mapped to $y$ by the Meyerson sketch.

Finally, notice that we store $O(\log_{1+\epsilon} \Delta)$ elements for each element in $\mathcal{C}$ (notice that $p$ is a constant). $\qquad\square$

## C.3    Suffix Sketch

The two augmentations before allow us to modify the Meyerson sketch in such a way that it can return an approximate solution for any suffix of the stream with length at most $w$. More precisely, we will show that we maintain a valid solution for the suffix with a cost comparable to the optimum of *the entire stream*.

Before proving the main result of this section, we need to add some additional bookkeeping to estimate the cost of the sketch inside the sliding window. The idea is similar to that used to maintain the approximate weight of a point.

**Lemma C.4.** *Let $\epsilon \in (0, 1)$ be a constant. Then using additional space $O\left(\log_{1+\epsilon}(M/m)\right)$ we can extend the Meyerson sketch $(\mathcal{C}, \mathsf{weight})$ with a function $\widehat{\mathsf{cost}_\mu} : [n] \to \mathbb{R}$ such that, for every time $\tau$ in the active window: $\mathsf{cost}_\mu(\tau) \leq \widehat{\mathsf{cost}_\mu}(\tau) \leq (1 + \epsilon)\mathsf{cost}_\mu(\tau)$, where $\mathsf{cost}_\mu(\tau)$ is the mapping cost for points inserted after time $\tau$*

*Proof.* The proof is very similar to the proof of Lemma C.1 and we include it here for completeness.

In the sketch, we maintain a sequence of cost $B = (b_1, b_2, \ldots)$ corresponding to different moving costs of $\mu$ and a sequence of times $G = (g_1, g_2, \ldots)$. The two sequences are initialized as empty. We preserve the invariant that for each time $g_j \in G$, the moving cost of $\mu$ for points inserted after time $g_j$ (inclusive) to the end of the stream is equal to $b_j$. To do so, when at time $i$, a point, $x_i$ is assigned to some center $y$ by $\mu$ (recall that the assignment is fixed), we increase by $d(x_i, y)^p$ all of the weights stored in $B$ and then we add a new weight initialized to $d(x_i, y)^p$ to $B$ and a new time equal to $i$ to $G$.

To reduce the size of the structure, we maintain only the significant changes in weights in $B$. More precisely, at any time, we delete $b_l$ and $g_l$ for any $l \in [2, \ldots, |B| - 1]$, if $b_{l-1} \leq (1 + \epsilon)b_{l+1}$. We also renumber the indices to be consecutive. Finally, we remove the $b_l$ (and the corresponding $g_l$) for which the weights are larger than $(1 + \epsilon)2^{p+7}M$.[9] Notice that at any time, for each $l$, either $b_l = b_{l+1} + 1$ or $b_l \leq (1 + \epsilon)b_{l+1}$. In fact, if $b_l$ and $b_{l+1}$ refer to consecutive assignments of points to the center, the first case is true. If $b_l$ and $b_{l+1}$ became consecutive after the removal of a point between them, the latter condition is true at the time of the removal, and is preserved by adding 1 to both elements.

Now, to compute the cost of mapping for every suffix, $\widehat{\mathsf{cost}_\mu}(\tau)$, we return the value $b_i$ in the $B$ array, where $i$ is the index of largest value smaller or equal to $\tau$ in $G$. Then $\mathsf{cost}_\mu(\tau) \leq \widehat{\mathsf{cost}_\mu}(\tau) \leq (1 + \epsilon)\mathsf{cost}_\mu(\tau)$. Finally, note that for each $l \in [|B| - 2]$, $b_l > (1 + \epsilon)b_{l+2}$, hence, the sequence is decreasing by a factor of $(1+\epsilon)$ every 2 steps, so that the total length is at most $O(\log_{1+\epsilon}(M/m))$. $\quad\square$

We note that the the algorithm defined previously can be easily modified to add the bookkeeping described in the previous lemmas, in particular we need only to modify the $ComputeMeyerson$ function.

**Algorithm 6** AugmentedMeyerson$(X, w, m, M, \Delta)$

---

1: **Input:** A sequence of points $X$, lower bound $m$ and upper bound $M$ to the optimum, and an upper bound $\Delta$ on the max distance.
2: **Output:** A data structure allowing to extract a $\epsilon$-consistent weighted instance of the substream $X_{[\tau,t]}$ with centers belonging to $X_{[\tau,t]}$.
3: Let parameters $p$ and $k$ be the parameters of the $k$-clustering problem.
4: Let $\gamma$ be the desired probability of success of the individual Meyerson algorithm.
5: Let $\epsilon$ be the parameter of the $\epsilon$-consistent mapping and $\epsilon$-replacement of centers as described in sections C.1 and C.2.
6: **for** $L^p \in \{m, 2m, 4m, 8m, \ldots, 2^{\lceil \log M/m \rceil} m\}$ **do**
7:     In parallel run $ComputeMeyerson(X, L^p, \alpha = 1/2, \gamma, \Delta, p, k)$ with bookkeeping for maintaining $\epsilon$-consistent mappings and $\epsilon$-replacement of centers as described in sections C.1 and C.2.
8: **end for**
9: Let $\ell$ be the smallest index (if it exists, otherwise it is an arbitrary index) in $\{m, 2m, 4m, 8m, \ldots, 2^{\lceil \log M/m \rceil} m\}$ for which the output $ComputeMeyerson$ called with $L_p = \ell$ has size smaller than $8k \log \gamma^{-1} (1 + \log \Delta) (2^{2p+3} + 1)$ and cost smaller than $2^{p+6} \ell$
10: Return the result of the index $\ell$ call for $ComputeMeyerson$ with the needed booking for allowing the operation $\mathsf{Suffix}_\tau$.

---

Now we can formally prove the main properties of our augmented Meyerson sketch.

**Lemma C.5** (Lemma 3.1 restated). *Let $w$ be the size of the sliding window, $\epsilon \in (0, 1)$ be a constant and $t$ the current time. Let $(X, \mathrm{d})$ be a metric space and fix $\gamma \in (0, 1)$. The augmented Meyerson algorithm computes an implicit mapping $\mu : X \to \mathcal{C}$, and an $\epsilon$-consistent weighted instance $(\mathcal{C}, \widehat{\mathrm{weight}})$ for all substreams $X_{[\tau,t]}$ with $\tau \geq t - w$ such that, with probability $1 - \gamma$, we have:*

$$|\mathcal{C}| \leq 2^{2p+8} k \log \gamma^{-1} \log \Delta \qquad \text{and}$$

$$f_p(X_{[\tau,t]}, \mathcal{C}) \leq 2^{2p+8} \mathrm{OPT}_p(X).$$

*The algorithm uses space $O(k \log \gamma^{-1} \log \Delta \log M/m (\log M + \log w + \log \Delta))$ and stores the cost of the consistent mapping, $f(X, \mu)$, and also a $1 + \epsilon$ approximation to the cost of the $\epsilon$-consistent mapping, $\widehat{f}(X_{[\tau,t]}, \mu)$.*

*Proof.* The main idea of the proof is to use our three bookkeeping tricks to provide a good sketch also for an arbitrary suffix of a stream. In particular our algorithm will run the Meyerson sketch with the additional bookkeeping as presented in Algorithm 5 and output a weighted instance and its cost using the additional bookkeeping.

Note that by Lemma B.1 we know that with probability $1 - \gamma$ the sets of center that we use in a Meyerson sketch are at most $2^{2p+8} k \log \gamma^{-1} \log \Delta$, our augmented Meyerson sketch change the center as describe in Lemma C.3 to guarantee that $\mathcal{C} \subseteq X_{[\tau,t]}$ but it never increases their number so the final number of center is still bounded by $2^{2p+8} k \log \gamma^{-1} \log \Delta$.

To bound the cost of the moving, we note that $f_p(X_{[\tau,t]}, \mathcal{C})$ is initially bounded by $2^{p+7} \mathrm{OPT}_p(X)$. Although in the augmented sketch we change the set of centers so the cost could increase, nevertheless for every point $y \in X_{[\tau,t]}$, we are guaranteed that there is a point in the final set of centers at distance at most $2^{p+1} d(y, \mu(y))^p$. This follows from Lemma C.3 and triangle inequality and because $2 + \epsilon < 4$. Finally the the factor $2^{2p+8}$ in the statements comes from multiplying $2^{p+1}$ with the approximation factor of Meyerson $2^{p+7}$.

Finally we notice that we can compute $\widehat{\mathrm{weight}}$ using the bookkeeping and the algorithm presented in Lemma C.1 and the $\widehat{\mathrm{cost}}_\mu(\tau)$ using bookkeeping and the algorithm presented in Lemma C.4.

The space bound follows from the space of the simple Meyerson sketch times the required space for bookkeeping. $\qquad \square$

For completeness, we provide a pseudocode for the AugmentedMeyerson (Algorithm 6) which is used in Algorithm 1. The AugmentedMeyerson corresponds to SimpleMeyerson (Algorithm 5) with the additional bookkeeping described in Section C. More precisely, AugmentedMeyerson stores for each sketch the data structures need for maintaining the estimate of the weights and centers described

in sections C.1 and C.2. Using these data structure the algorithm we can extract the weighted set $\mathsf{Suffix}_\tau(S) = (\mathcal{C}, \widehat{\mathsf{weight}})$ for a given $\tau$ which represents an $\epsilon$-consistent weighted instance of the substream $X_{[\tau,t]}$ with centers $C$ belonging to $X_{[\tau,t]}$.

# D  Omitted Proofs from Sliding Windows subsection

**Lemma D.1** (Lemma 3.2 restated). *Using an approximation algorithm* $\mathsf{ALG}$, *from the augmented Meyerson sketch* $S(Z)$, *with probability* $\geq 1 - \gamma$, *we can output a solution* $\mathsf{ALG}(S(Z))$ *and an estimate* $\widehat{f}_p(S(Z), \mathsf{ALG}(S(Z)))$ *of its cost s.t.*

$$f_p(Z, \mathsf{ALG}(S(Z))) \leq \widehat{f}_p(S(Z), \mathsf{ALG}(S(Z)))$$
$$\leq \beta(\rho) f_p(Z, \mathrm{OPT}(Z))$$

*for a constant* $\beta(\rho) \leq 2^{3p+6}\rho$ *depending only the approximation factor* $\rho$ *of* $\mathsf{ALG}$.

*Proof.* Given the sketch $S(Z)$ computed with Algorithm 5, we obtain a set of centers $Y(Z)$ and theirs weights $\mathsf{weight}(Z)$, and a cost of the mapping $\mathsf{cost}_\mu(Z)$ (as well as an implicitly defined consistent mapping $\mu$).

So we can construct a weighted instance $(Y(Z), \mathsf{weight}(Z))$ and solve it approximately with $\mathsf{ALG}$ to obtain centers $\mathcal{C} = \mathsf{ALG}(S(Z))$ which are given in output. We now bound the cost of $\mathcal{C}$ over $Z$.

As we have shown in the proof of Lemma 2.1, we have that $f_p(Z, \mathcal{C}) \leq 2^{p-1}(\mathsf{cost}_\mu(Z) + f_p(Y(Z), \mathsf{weight}(Z), \mathcal{C}))$, where $f_p(Y(Z), \mathsf{weight}(Z), \mathcal{C})$ is the cost of the solution $\mathcal{C}$ over the weighted instance. We also know that the weighted instance has an optimum cost of $\leq 2^{2p-1}(\mathsf{cost}_\mu(Z) + \mathrm{OPT}_p(Z))$. So applying an $\rho$-approximate algorithm we get

$$f_p(Y(Z), \mathsf{weight}(Z), \mathcal{C}) \leq \rho 2^{2p-1}(\mathsf{cost}_\mu(Z) + \mathrm{OPT}_p(Z))$$

Finally we get that $f_p(Z, \mathcal{C}) \leq \rho 2^{2p}(\mathsf{cost}_\mu(Z) + \mathrm{OPT}_p(Z))$ which by Lemma B.3 gives $f_p(Z, \mathcal{C}) \leq 2^{3p+6}\rho \, \mathrm{OPT}_p(Z)$, with probability $\geq 1 - \gamma$, so the theorem holds with $\beta(\rho) := 2^{3p+6}\rho$.

Finally, notice that the value $\widehat{f}_p(S(Z), \mathsf{ALG}(S(Z))) := 2^{p-1}(\mathsf{cost}_\mu(Z) + f_p(Y(Z), \mathsf{weight}(Z), \mathcal{C}))$, can be computed from $S(Z)$ (using $\mathsf{ALG}$) so we can output it as an estimate of the cost. Notice, that this is valid a lowerbound of the actual cost on $Z$ of the solution, and it is $\leq \beta(\rho) \mathrm{OPT}_p(Z)$. $\square$

Now we prove some basic properties of Algorithm 1] that we will use in other prove.

**Lemma D.2** (Invariants of Algorithm 1). *For a set* $A_\lambda$, *let* $A_\lambda^+$ *be the set* $A_\lambda$ *together with the first element of the corresponding* $B_\lambda$. *The following are invariants maintained by the algorithm:* (i) $A_\lambda$ *and* $B_\lambda$ *are two disjoint consecutive substreams of* $X$, (ii) $A_\lambda$ *precedes* $B_\lambda$, *and* (iii) $B_\lambda$ *ends with the current time* $t$ *and always contains the last element of the stream.* (iv) *When* $|A_\lambda| \geq 1$: $\widehat{f}_p(S(A_\lambda^+), \mathsf{ALG}(S(A_\lambda^+))) > \lambda$. (v) *When* $|A_\lambda| \geq 2$: $\widehat{f}_p(S(A_\lambda), \mathsf{ALG}(S(A_\lambda))) \leq \lambda$. (vi) *When* $|B_\lambda| \geq 2$: $\widehat{f}_p(S(B_\lambda), \mathsf{ALG}(S(B_\lambda))) \leq \lambda$

*Proof.* The first three points follow from construction. The fourth follows from the fact that $A_\lambda$ is set to a non-empty sketch only when the value associated with $B_\lambda \cup \{x\}$ is $> \lambda$ and this property is maintained thereafter. Equation (v) follows from the fact that $|A_\lambda| \geq 2$ implies that $A_\lambda$ has been set as a copy of a prior $B_\lambda$ which was not a singleton, and this happens only if the value of the cost of the sketch associated with $B_\lambda$ is $< \lambda$. Equation (vi) follows from the same property. $\square$

**Lemma D.3** (Composition with a Suffix of stream, restatement of Lemma 3.3). *Given two substreams* $A, B$ *(with possibly* $B = \emptyset$*) and a time* $\tau$ *in* $A$, *let* $\mathsf{ALG}$ *be a constant approximation algorithm for the $k$-clustering problem. Then if* $\mathrm{OPT}_p(A) \leq O(\mathrm{OPT}_p(A_\tau \cup B))$, *then, with probability* $\geq 1 - O(\gamma)$, *we have* $f_p(A_\tau \cup B, \mathsf{ALG}(\mathsf{Suffix}_\tau(S(A)) \cup S(B))) \leq O(\mathrm{OPT}_p(A_\tau \cup B))$.

*Proof.* Let $\mu_A$ be the mapping consistent with the sketch $S(A)$ and $\mu_B$ be the mapping consistent with the sketch $S(B)$. Note that from this two mappings we can construct a third mapping $\mu$ such that $\mu(x) = \mu_A(x)$ if $x \in A$ and $\mu(x) = \mu_B(x)$ if $x \in B$(note that input points are in

general position so they are all in distinct position). The mapping $\mu$ is now a valid mapping for $A_\tau \cup B$, furthermore we can note that the number of points mapped to a point $y$ by the mapping is equal to $|\{x \in A : \mu(x) = y\}| + |\{x \in B : \mu(x) = y\}|$, so by definition of the sketch $\mathsf{Suffix}_\tau(S(A)) \cup S(B)$ and by the fact that $\mathsf{Suffix}_\tau(S(A))$ is $\epsilon$-consistent and $S(B)$ is consistent we obtain that $\mathsf{Suffix}_\tau(S(A)) \cup S(B)$ is $\epsilon$-consistent with $\mu$.

Note from Lemma B.1 we know that the moving cost of $\mu_A$ on $A$ is in $O(\mathrm{OPT}_p(A))$ with probability $\geq 1 - \gamma$. Similarly the moving cost of $\mu_B$ on $B$ is in $O(\mathrm{OPT}_p(B))$ with probability $\geq 1 - \gamma$, so the moving cost of the function $\mu$ on $A \cup B$ is $O(\mathrm{OPT}_p(A) + \mathrm{OPT}_p(B)) \in O(\mathrm{OPT}_p(A_\tau \cup B))$ by the hypothesis in our Lemma with probability $\geq 1 - 2\gamma$. Furthermore also the moving cost of $\mu$ on $A_\tau \cup B$ is in $O(\mathrm{OPT}_p(A_\tau \cup B))$.

Now, let $\mathcal{C}_{\mathsf{ALG}}$ be the centers selected by a constant approximation algorithm on $\mathsf{Suffix}_\tau(S(A)) \cup S(B)$. From generalized triangle inequality we get $f_p(A_\tau \cup B, \mathcal{C}_{\mathsf{ALG}}) = \sum_{x \in A_\tau \cup B}$ $d(x, \mathcal{C}_{\mathsf{ALG}})^p \leq \sum_{x \in A_\tau \cup B} 2^{p-1}(d(x, \mu(x))^p + d(\mu(x), \mathcal{C}_{\mathsf{ALG}})^p) \in O(\mathrm{OPT}_p(A_\tau \cup B) + \sum_{x \in A_\tau \cup B} d(\mu(x), \mathcal{C}_{\mathsf{ALG}})^p)$. Now from Lemma 2.1 we know that the instance obtained by applying the mapping $\mu$ to $A_\tau \cup B$ has optimal solution $\mathcal{C}_\mu$ with cost in $O(\mathrm{OPT}_p(A_\tau \cup B))$. Unfortunately we do not apply the algorithm directly on an instance consistent with the mapping, nevertheless by $\epsilon$-consistency we know that the cost of the set of centers $\mathcal{C}_\mu$ for the instance $\mathsf{Suffix}_\tau(S(A)) \cup S(B)$ is most $(1 + \epsilon)$ times the cost for the instance obtained by the mapping. So also the optimal solution of $\mathsf{Suffix}_\tau(S(A)) \cup S(B)$ has cost in $O(\mathrm{OPT}_p(A_\tau \cup B))$. So also $\mathcal{C}_{\mathsf{ALG}}$ have cost in $O(\mathrm{OPT}_p(A_\tau \cup B))$ because $\mathsf{ALG}$ is a constant approximation algorithm. The Lemma follows by noticing that by definition $\epsilon$-consistent $\sum_{x \in A_\tau \cup B} d(\mu(x), \mathcal{C}_{\mathsf{ALG}})^p \leq f_p(\mathsf{Suffix}_\tau(S(A)) \cup S(B), \mathcal{C}_{\mathsf{ALG}})$. $\qquad\square$

**Theorem D.4** (Theorem 3.4 restated). *With probability $1 - \gamma$, Algorithm 2, outputs an $O(1)$-approximation for the sliding window $k$-clustering problem using space: $O\big(k \log(\Delta)(\log(\Delta) + \log(w) + \log(M)) \log^2(M/m) \log(\gamma^{-1} \log(M/m))\big)$ and total update time $O(T(k \log(\Delta), k) \log^2(M/m) \log(\gamma^{-1} \log(M/m)) (\log(\Delta) + \log(w) + \log(M))$.*

*Proof.* Notice that at any point in time we run at most $2 \log_{1+\delta}(M/m)$ Augmented Meyerson sketches. We can set each of them to use as probability of error bound $\frac{\gamma}{2 \log_{1+\delta}(M/m)}$ to have total probability of any of them failing $\leq 1 - \gamma$. We continue the analysis on assuming we are in the case all current sketches did not fail.

Consider the sketches maintained by the algorithm. If for some $\lambda$, the active window W is identical to the interval $B_\lambda$, we return $\mathsf{ALG}(S_2)$ as the solution. Since $S_2$ is a Meyerson sketch we have that the weighted instance computed by the sketch is consistent with a mapping of cost at most a constant factor larger than the cost of the optimal solution in the active window and so by Lemma 2.1 we have that $\mathsf{ALG}(S_2)$ is a constant approximation. (Note that this is independent of the value of $\lambda$, in fact $\lambda$ is not used by the Meyerson sketch that independently tries several possible lower bounds for the cost of the solution)

Otherwise, we find the maximum $\lambda' \in \Lambda$ for which $A_{\lambda'}$ is not empty and fully contained in the current active window. As long as $|W| > 1$, for the sketches associated with the smallest $\lambda$, we have $A_m \neq \emptyset$, which guarantees that such a $\lambda'$ exists. Here $m$ is the lower bound on the optimum value as defined in Section 2. Furthermore, let $\lambda^* = \lambda'(1 + \delta)$. Note that the fact that for the largest $\lambda$, all elements are contained in $B_\lambda$, this guarantees that such a $\lambda^*$ exists in the set of thresholds $\Lambda$ as well.

We now show that $\lambda' < O(\mathrm{OPT}(W))$, which implies that $\lambda^* < (1 + \delta)O(\mathrm{OPT}(W))$. In fact, suppose $\lambda'$ is such that $A_{\lambda'}$ is non-empty. We know that $A_{\lambda'} \subseteq W$. Let $A_{\lambda'}^+$ be $A_{\lambda'}$ plus the first element of $B_\lambda$, by definition of the algorithm, we have that $\lambda' < \widehat{f}_p(S_{A_{\lambda'}^+}, \mathsf{ALG}(S(A_{\lambda'}^+)))$ where the inequality follows from the invariant (iv) of Lemma D.2, but now $\widehat{f}_p(S_{A_{\lambda'}^+}, \mathsf{ALG}(S(A_{\lambda'}^+)))$ is the cost of a solution computed on the Meyerson sketch for $A_{\lambda'}^+$ for $A_{\lambda'}^+$ so using the fact that the Meyerson sketch provides a constant approximation we get $\lambda' < \widehat{f}_p(S_{A_{\lambda'}^+}, \mathsf{ALG}(S(A_{\lambda'}^+))) \leq O(\mathrm{OPT}(A_{\lambda'}^+))$ but now we can notice that by adding points to any set $X$ we can decrease the cost of the solution by at most a factor $2^p$(for a formal proof of this fact look at Lemma A.2) so we get $\lambda' < \widehat{f}_p(S_{A_{\lambda'}^+}, \mathsf{ALG}(S(A_{\lambda'}^+))) \leq O(\mathrm{OPT}(A_{\lambda'}^+)) \leq 2^p O(\mathrm{OPT}(W)) \in O(\mathrm{OPT}(W))$.

So, we can then assume that $\lambda^* < (1 + \delta)O(\mathrm{OPT}(W))$. By definition of $\lambda^*$, $W \subseteq A_{\lambda^*} \cup B_{\lambda^*}$. There are three cases to consider.

- W is a strict suffix of $B_{\lambda^*}$. In this case, the W is equal to a suffix of $B_{\lambda^*}$ starting at position $\tau$ for some $\tau$. Notice that $|B_{\lambda^*}| \geq 2$ so we know that the sketch over $B_{\lambda^*}$ has cost $\leq \lambda^*$. so we can apply the Lemma 3.3 to compose the suffix of $B_{\lambda^*}$ with the empty sketch.

- W $= B_{\lambda^*}$, this ensures to a constant approximation.

- W intersects with $A_{\lambda^*}$. We start by computing $\mathsf{Suffix}_t(S_1)$, and then we compute $\mathsf{ALG}(\mathsf{Suffix}_t(S_1) \cup S_2)$. Again we know that the $A_{\lambda^*}$ has a sketch of small cost so we can apply the composition Lemma 3.3.

Notice that the space bound comes from the total number of points in all sketches used. Notice also that the number of distance function evaluations (and thus the update time) to update the Meyerson sketches per each point in the stream is bounded by the number of points in all sketches (we do one evaluation for each center). The total update time depends also on the complexity of $\mathsf{ALG}$, but we only run it on instances of $k \log(\Delta)$ points, so the total update time is $O\big(T(k \log(\Delta), k) \log^2(M/m) \log(\gamma^{-1} \log(M/m))(\log(\Delta) + \log(w) + \log(M))\big)$. Notice also that at the last step, to return a solution, after the updates are done, we only require solving a constant number of $k$-clustering instances with $\mathsf{ALG}$, again each of them of size $O(k \log(\Delta))$. $\qquad\square$

# E  Empirical analysis

## E.1  Additional information on the experimental setup

**Baselines.** We consider the following baselines. **Off-Line K-Means++**: We use $k$-means++ over the entire window as a proxy for the optimum, since the latter is NP-hard to compute. At every insertion, we report the best solution over 10 runs of $k$-means++ on the window. Observe that this is inefficient as it requires $\Omega(w)$ space and $\Omega(kw)$ cost per update. Specifically, we run $k$-means++ 10 times and report the best solution obtained. To compare the cost of our algorithm to the optimum, we run $k$-means++ every time a new element is inserted. While this gives the best cost, this is obviously inefficient: such an approach requires $\Omega(w)$ space and $\Omega(kw)$ cost per update. **Sampling**: We maintain a random sample of points from the active window, and then run $k$-means++ on the sample. This allows us to evaluate the performance of a baseline, at the same space cost of our algorithm. **SODA16**: We also evaluated the only previously published algorithm for this setting in [17]. We note that we made some practical modifications to further improve the performance of our algorithm which we now describe.

For our algorithm we used a standard optimization from the literature [44]: running one copy of the Meyerson sketch instead of the $O(\log(1/\gamma))$ copies that are needed for high probability statements. We also developed a lazy evaluation of the cost of the Meyerson sketch that saves update time. For the constant $\alpha$ in the Meyerson algorithm we use .5, as shown in the pseudo-code.

Second, as we focus on the cost of the solution, instead of the centers output, we ignore the presence of the center in the window (i.e., we do not use the method to output centers in the window). This is consistent with the problem addressed by [17] in which they do not restrict the centers to be in the sliding window. This way we allow a fair comparison of our results with this baseline.

Third, similarly to [44], in the implementation of the update function, the cost of the sketch is estimated through a lazy evaluation in which the cost is not recomputed from the last time unless at least one new center is added or the number of points or total cost associated with a center is increased by a factor $(1 + \eta)$ from the last full evaluation (we use $\eta = 0.05$); we stop a Meyerson sketch only when the size bound is exceeded, not the cost bound.

Finally, to output a solution, instead of the specific pair of summaries described in our algorithm, we make best-effort use of all available pairs of sketches associated with all guesses of the optimum and return the best one with lowest estimated cost–this also makes the algorithm more resistant to error in estimating the lower and upper bounds of the cost. Our empirical analysis shows that these techniques speed up the computation significantly while retaining very strong approximation factors in real dataset, as we show in this section.

| Dataset | $k$ | Space Decr. Factor | Speed-Up Factor | Cost (ratio) |
|---------|-----|--------------------|-----------------|--------------|
| COVER | 4 | 5.23 | 10.88 | 99.5% |
| | 8 | 4.86 | 10.30 | 99.9% |
| | 10 | 4.86 | 10.36 | 97.6% |
| | 16 | 6.04 | 13.10 | 95.1% |
| | 20 | 7.09 | 16.45 | 94.3% |
| SHUTTLE | 4 | 5.07 | 9.09 | 106.8% |
| | 8 | 6.09 | 7.89 | 102.4% |
| | 10 | 6.28 | 7.79 | 103.2% |
| | 16 | 15.32 | 15.14 | 118.9% |
| | 20 | 13.30 | 12.32 | 143.7% |

Table 3: Decrease of space use, decrease in (speed-up) and ratio of mean cost of the solutions of our algorithm vs the SODA16 baseline. Space decrease and the speed-up factors are multiplicative, the cost column reports the ratio of the cost of our algorithm over that of the SODA16 algorithm.

**Upper and lower bounds on cost.** Our algorithms described before require an estimate of the lower and upper bounds for the cost to initialize the thresholds used in the summaries. In the Appendix F we show that such assumptions can be removed at the expenses of a more complicated algorithm. In our experiments instead of removing the assumptions we have implemented the simpler algorithms described in the paper and we used an heuristic to estimate the bounds $m, M$ in input to the algorithms. Here we describe the heuristic, which empirically results in strong approximation results and efficiency: sample a number of window-sized substreams from (a prefix) of the stream and compute an approximate solution for each window with any algorithm (e.g., kmeans++ or an insertion-only stream algorithm); then use the min and max cost found (with a slack) for the bounds.

More precisely, let $m', M'$ be the minimum and maximum cost observed in the samples and let $\mu, \sigma$ be the empirical mean and standard deviation of the samples' costs. To estimate $m, M$ we use $m := \max(\frac{m'}{3}, \mu - 3\sigma)$ and $M := \max(3M, \mu + 3\sigma)$. We use 10 samples in our experiments for the estimation. Note that the $\max$ with $\frac{m'}{3}$ is used to ensure that the lower bound is set to be $> 0$. We use the same technique in the evaluation of the SODA16 baseline [17]. For the $\Delta$ input of our algorithms we use the $M'$ obtained from the heuristic.

**Implementation details for [17].** To the best of our knowledge, this algorithm has not been evaluated empirically. In implementing this algorithm we used the same speed up techniques used for our algorithm: we used one instead of logarithmic many Meyerson sketches, we used the same implementation of the Meyerson sketch used for our algorithm for fairness of comparison. We also provide the same upper and lower bounds on the optimum used for our algorithm. Finally, we set the grid width parameter $\delta$ of the algorithm, as the equivalent $\delta$ parameter of our algorithm. We set similarly the $\beta$ parameter [17].

## E.2 More detailed comparison with previous work

Here we report a more detailed analysis in Table 3 where we show that our algorithm uses up to $15x$ less space, does up to $16x$ fewer distance calculations and achieves a solution up to 7% cheaper. Notice how the improvements over the baseline grow with $k$ as predicted by theory. We observe that for $k = 16$ on the SODA16 baseline always has roughly the same space or more that that required to store the entire sliding window making the sliding window algorithm trivial.

Notice how our algorithm result in substantial improvements even for small values of $k$.

## E.3 Additional experiments on cost of the solution

Observe that our algorithm closely tracks that of the optimum, even as the cost fluctuates up and down. Notice how the random sampling baseline presents a particularly high approximation factor in COVERTYPE (Fig. 3(a)) during timesteps when there is a shift in the cost magnitude; this confirms the importance of sliding window algorithms to detect correctly (and re-cluster appropriately) shifts in data promptly. Similar results sre observed in all datasets Figure 3. To capture this analytically, we show the median relative error in Table 4. While our algorithm has a median error within 10% of KM++, the median error of the sampling baseline is 30-40% higher than KM++. Notice how the size of the sliding window does not affect substantially the experimental results.

| Dataset | $W$ | Sketch | Sampling | Reduction |
|---|---|---|---|---|
| COVER | 10000 | 1.09 (0.14) | 1.29 (0.31) | $-15.9\%$ |
| | 20000 | 1.05 (0.12) | 1.31 (0.30) | $-19.4\%$ |
| | 40000 | 1.06 (0.13) | 1.41 (0.44) | $-25.0\%$ |
| SHUTTLE | 10000 | 1.11 (0.16) | 1.48 (0.25) | $-24.8\%$ |
| | 20000 | 1.09 (0.14) | 1.32 (0.14) | $-17.4\%$ |
| | 40000 | 1.07 (0.11) | 1.26 (0.13) | $-15.0\%$ |
| SKIN | 10000 | 1.13 (0.19) | 1.40 (0.40) | $-19.7\%$ |
| | 20000 | 1.10 (0.17) | 1.34 (0.32) | $-17.8\%$ |
| | 40000 | 1.09 (0.14) | 1.31 (0.28) | $-16.4\%$ |

Table 4: Mean cost ratio of our algorithm (std-dev in parenthesis) and of the sampling baseline over the $k$-means++ gold standard for $k = 10$, $\delta = 0.2$

(a) COVERTYPE       (b) SHUTTLE       (c) SKINTYPE

Figure 3: Cost of the solution obtained by our algorithm (Sketch) and the two baselines for $k = 20$, $|W| = 40,000$ and $\delta = 0.2$. Notice that our algorithm's cost is close to that of the off-line algorithm and significantly better than the random sampling baseline

## E.4 Additional experiments on update time and space tradeoff

Additional experiments are presented in Table 5 where we observe substantial savings w.r.t. the naive method. Notice how the savings increase when the size $w$ grows.

## E.5 Recovering ground-truth clusters.

In this section we evaluate the accuracy of the clusters produced by our algorithm on a dataset with ground-truth clusters. Note that the ground truth is for all the all set so in our setting we use the cluster assignments of the original ground truth restricted to the particular sliding window. We compared the offline KM++ baseline (KM++), our algorithm (Sketch), and the same-space (Sampling) baseline. We do not evaluate the SODA16 baseline because it is dominated by KM++ in space use and solution quality. We use the well known V-Measure accuracy definition for clustering [52].

We report in Table 6 the results for our empirical evaluation of the accuracy of our algorithm on identifying the ground truth clusters in synthetic datasets.

We observe that our algorithm always dominates the Sampling baseline for all parameters and datasets and it has results comparable to the gold standard offline algorithm while using significantly less space and running time.

| $W$ | $k$ | Space | Time |
|---|---|---|---|
| 20000 | 10 | 7.2% (0.75%) | 1.2% (0.68%) |
| | 20 | 13.72% (2.06%) | 2.6% (1.5%) |
| | 40 | 21.3% (3.4%) | 3.5% (2.8%) |
| 40000 | 10 | 3.5% (0.39%) | 0.45% (0.29%) |
| | 20 | 6.5% (0.87%) | 0.93% (0.63%) |
| | 40 | 11.3% (1.74%) | 1.58% (1.23%) |

Table 5: Max percentage of sliding window stored (Space) and median percentage of time (Time) vs. one run of $k$-means++ (stdddev in parantesis).

| Dataset | W | Off-line | Sketch | Sampling |
|---|---|---|---|---|
| s1 | 2000 | 0.958 (0.016) | 0.964 (0.019) | 0.927 (0.026) |
|    | 3000 | 0.965 (0.015) | 0.969 (0.015) | 0.926 (0.025) |
|    | 4000 | 0.969 (0.012) | 0.969 (0.017) | 0.933 (0.030) |
| s2 | 2000 | 0.901 (0.018) | 0.903 (0.015) | 0.856 (0.026) |
|    | 3000 | 0.889 (0.014) | 0.903 (0.014) | 0.860 (0.028) |
|    | 4000 | 0.901 (0.014) | 0.903 (0.021) | 0.851 (0.025) |
| s3 | 2000 | 0.741 (0.018) | 0.748 (0.013) | 0.717 (0.019) |
|    | 3000 | 0.740 (0.017) | 0.745 (0.011) | 0.717 (0.015) |
|    | 4000 | 0.735 (0.018) | 0.746 (0.017) | 0.712 (0.013) |
| s4 | 2000 | 0.678 (0.015) | 0.677 (0.017) | 0.665 (0.019) |
|    | 3000 | 0.678 (0.017) | 0.674 (0.015) | 0.659 (0.019) |
|    | 4000 | 0.676 (0.009) | 0.675 (0.013) | 0.666 (0.013) |

Table 6: Average V-Measure of the cluster using the $k$-means++, sampling baselines and, our algorithm as compared to the ground truth clusters in synthetic datasets (stddev in paranthesis). We use $\delta = 0.2$.

| Dataset | $W$ | Sketch | Sampling | Reduction |
|---|---|---|---|---|
| COVER | 10000 | 1.64 | 4.09 | $-60.0\%$ |
|       | 20000 | 1.49 | 4.05 | $-63.1\%$ |
|       | 40000 | 1.62 | 3.79 | $-57.2\%$ |
| SHUTTLE | 10000 | 2.03 | 2.53 | $-20.0\%$ |
|         | 20000 | 1.69 | 1.80 | $-6.3\%$ |
|         | 40000 | 1.45 | 1.75 | $-17.00\%$ |
| SKIN | 10000 | 2.28 | 4.16 | $-45.3\%$ |
|      | 20000 | 1.19 | 4.13 | $-54.2\%$ |
|      | 40000 | 1.76 | 3.27 | $-46.3\%$ |

Table 7: Maximum cost ratio of our algorithm and the sampling baseline over the $k$-means++ gold standard for $k = 10$, $\delta = 0.2$

### E.6 Additional empirical results on comparisons with ths baselines

**Max cost ratios vs gold standard**   Our improvement over this baseline is even larger if we look at max error (Table 7) over the entire stream. Notice that, in these experiments, our algorithm always has less than 2.2 cost factor, while the sampling baseline can return costs that are $> 4$x higher than the gold standard, thus resulting in a reduction of up to $-60\%$ in cost. Note also that the empirical approximation ratios are much better than what the worst case theoretical analysis pessimistically predicts.

We report in Table 8 additional results.

**Comparison of costs with baseline**   We now analyze quantitatively the cost of the solutions obtained by our algorithm and the other baselines. Table 4 reports the median ratio (over all timesteps evaluated) between the cost obtained by our algorithm and the $k$-means++ gold standard (column Sketch) as well as the median ratio of the sampling baseline over the gold standard (Sampling) for $k = 10$, $\delta = 0.2$, and various $|W|$'s. Notice how our algorithm has a median approximation factor (w.r.t. the gold standard) of $5 - 13\%$ higher cost and a reduction of cost w.r.t. the sampling baseline of up to $-25\%$. An approximation factor of $< 1.2$ is significantly better than the pessimistic worst-case theoretical analysis and shows that our algorithm is highly precise in practice.

**Fraction of window stored**   Figure 5 we show the percent of the sliding window data points stored as a function of the desired number of clusters, $k$, for the COVERTYPE dataset and $W = 40{,}000$ (other results are similar). Observe that we always store a small fraction of the dataset, and that the bounds grow linearly with $k$, as suggested by the theory. In Finally, In Figure 4 we show the average number of points stored per individual sketch. Notice how the number of points stored is mostly unaffected by $w$, thus confirming that our algorithm can scale to large window sizes.

Figure 4: Number of items per sketch stored by our algorithm.

| Dataset | $W$ | $k$ | Space | Time |
|---------|------|-----|-------|------|
| COVER | 10000 | 5 | 7.24 | 1.57 |
| | | 10 | 14.43 | 3.12 |
| | | 20 | 28.33 | 6.56 |
| | | 40 | 37.22 | 6.33 |
| | 20000 | 5 | 3.66 | 0.62 |
| | | 10 | 7.28 | 1.22 |
| | | 20 | 13.72 | 2.62 |
| | | 40 | 21.08 | 3.51 |
| | 40000 | 5 | 1.83 | 0.24 |
| | | 10 | 3.51 | 0.46 |
| | | 20 | 6.53 | 0.93 |
| | | 40 | 11.34 | 1.59 |
| SHUTTLE | 10000 | 5 | 3.86 | 0.57 |
| | | 10 | 5.77 | 0.88 |
| | | 20 | 8.23 | 1.60 |
| | | 40 | 4.81 | 0.58 |
| | 20000 | 5 | 1.60 | 0.18 |
| | | 10 | 1.72 | 0.17 |
| | | 20 | 2.22 | 0.20 |
| | | 40 | 2.22 | 0.18 |
| | 40000 | 5 | 0.72 | 0.09 |
| | | 10 | 0.69 | 0.05 |
| | | 20 | 0.90 | 0.07 |
| | | 40 | 0.94 | 0.05 |
| SKIN | 10000 | 5 | 7.08 | 1.61 |
| | | 10 | 9.81 | 1.18 |
| | | 20 | 7.09 | 0.43 |
| | | 40 | 9.14 | 0.29 |
| | 20000 | 5 | 3.26 | 0.62 |
| | | 10 | 4.15 | 0.51 |
| | | 20 | 3.38 | 0.22 |
| | | 40 | 4.64 | 0.16 |
| | 40000 | 5 | 1.67 | 0.26 |
| | | 10 | 2.46 | 0.28 |
| | | 20 | 1.85 | 0.12 |
| | | 40 | 2.10 | 0.08 |

Table 8: Max percentage of the items stored w.r.t. the sliding window (Space) and median percentage of distance evaluation vs. one run of $k$-means++ for $\delta = 0.2$

Figure 5: Space usage of our algorithm for different values of $k$ and $\delta$ and $W = 40,000$ in COVER-TYPE. Space usage is reported as percent of he window points stored .

### E.7  Experiments using a different $p$

All the experiments we have reported are for k-means ($p = 2$). We observe that our implementation easily adapts to any $p$ by just changing the underlying $k$-clustering solver. For sake of completeness, we also ran some experiments with the $k$-median objective, $p = 1$. All the results we observe are in line with that of the k-means case ($p = 2$). For instance, replicating the clustering accuracy experiment using $p = 1$ (See Table 6 for the results for $p = 2$) we obtain the following V-Measure accuracy for the offline algorithm, our stream algorithm and the sampling baseline (same space), $81.8\%$, $79.9\%$, $78.5\%$, respectively. This confirming the same trend observed for k-means. We observe the same trend in terms of speedups and space savings w.r.t. the gold-standard baseline.

## F   Appendix: Relaxing the assumptions on the input

In the paper we made the following assumptions for the sake of exposition:

- the distances between points are normalized to lie between 1 and $\Delta$,
- the points are distinct, and
- we have access to a lower bound $m$ and upper bound $M$ of the cost of the optimal solution in any sliding window.

We can relax these assumptions as follows.

### F.1   Relaxing the distinct points' assumption.

Notice that we made the assumption that the points in $X$ are distinct only for sake of simplifying the notation of the mapping function from points to centers (we want to define a function from a point). In general, points can be repeated in the stream and without any loss of generality one can define a function (implicitly) on the points assuming their arrival time make them distinct entities (note that we do not need to store the mapping which is not used by the algorithm, this is only for sake of analysis).

### F.2   Relaxing the assumption having distances in $[1, \Delta]$ in Meyerson algorithm

Notice that in the Meyerson algorithm (Algorithm 3) we assume that the min distance of two non-coincident points is 1 and we assume to know the maximum distance $\Delta$. The latter value is used to sample points with probability depending on $\log(\Delta)$. In reality we do not need such assumptions. Notice that the $\log(\Delta)$ factor is needed to perform a union bound over the number of exponentially growing balls around a center that starts at distance given by the average cost of a point in a cluster. It is possible to observe that the maximum distance from a point in a cluster $C$ to the center, is a most $|C|$ times the average distance of points to the center in the cluster. So if we assume that the algorithm

is run on a stream of size at most $O(w)$ we can use $\log(w)$ instead of $\log(\Delta)$ in the algorithm, and ignore the assumption on the min distance of points. In the Section F.5 we show that we never need to compute the sketches on streams of length larger than $2w$ so this we can replace $\log(\Delta)$ with $\log(2w)$ in Algorithm 3.

### F.3 Estimating the min cost $m$

The final assumption is knowing a lower bound on the minimum cost $m$ and an upper bound on cost, $M$ for defining the grid containing the guesses of the optimum. To have a lower bound on $m$, we maintain the most recent $k + 1$ distinct points at all times in a set $S$. Whenever a new point $x$ arrives, we check to see if we can replace one of the $k + 1$ points in $S$ with $x$. For instance if $x \notin S$, we add $x$ to $S$ and remove the most stale (oldest) point in $S$. If an identical point $y = x$ already exists in set $S$, we just replace $y$ with this new copy $x$. This way, we always keep the most recent distinct $k + 1$ points in $S$. If some of the points in $S$ lie outside the current active window $W$, the total number of distinct points in the active window does not exceed $k$. In this case, the optimum clustering has cost zero and we have access to the $\leq k$ centers yielding this optimum clustering. Otherwise, all points in $S$ belong to the active window $W$. Therefore the minimum cost of clustering $S$ lower bounds the optimum clustering cost of the whole window $W$. Notice that the minimum cost of clustering $S$ is simply the minimum pairwise distance between points in $S$. This is a valid lower bound on $m$.

As the sliding window shifts, the lower bound on $m$ might increase or decrease. Whenever, the lower bound increases, in algorithm $SimpleMeyerson(X_t)$, we need to discard some runs of $ComputeMeyerson(X_t, L^p)$. These are instances where $L^p$ is smaller than the new lower bound on $m$ and therefore not relevant anymore. Discarding those runs is easy to do.

However, when the lower bound on $m$ decreases, we need to instantiate new runs of $ComputeMeyerson(X_t, L^p)$ for some smaller values of $L^p$. But these new runs are only valid for sliding windows that consist of only the current set of points in $S$. Older versions of $S$ were providing higher lower bounds on $m$ and therefore we do not need to do anything for those older sliding windows. So the only sliding window for which we need to run new instances of $ComputeMeyerson(X_t, L^p)$ is the sliding window starting at time when $S$ begins, which consists of only distinct points in $S$ so far. Notice that we can easily keep the multiplicities of the coincident points for each point in $S$. Therefore we have all the information we need to run the new instances of $ComputeMeyerson(X_t, L^p)$ for the new smaller values of $L^p$ simulating the algorithm on a arbitrary ordering of points (with multiplicities) from $S$.

### F.4 Estimating the max cost $M$

Now we will talk about maintaining an upper bound $M$ on the optimum cost. The number of points in the window is $w$, and the optimum cost cannot be more than $w$ times the diameter of the points in the sliding window. Cohen-Addad et al. [26] design an algorithm that approximates the diameter of points in a sliding window with a constant $3 + \epsilon$ approximation guarantee and using $O(\log(\Delta)/\epsilon)$ memory. Therefore, we can always use this dynamically changing upper bound on $M$. Whenever our upper bound decreases, we need to discard some runs of $ComputeMeyerson(X_t, L^p)$ for values of $L^p$ that are greater than the new upper bound on $M$ which is easy to do.

However, when the upper bound on $M$ increases, there are new values of $L^p$ for which we need to run $ComputeMeyerson(X_t, L^p)$. The challenge is to run these sketches retrospectively. We do not have access to the whole active window $W$. Let $L^{old}$ be the highest value of $L^p$ we had before we increased our upper bound on $M$. For any new value of $L^p$ that exceeds our previous upper bound on $M$ (let's call this new value $L^{new}$), we initialize the sketch with the sketch we have maintained so far for $L^{old}$. In other words, we use the closest value of $L^p$ to initialize all new sketches.

In the sampling procedure, $L^p$ is the denominator in the sampling probability. So reusing the sketch with lower $L^p$ means that the previous points have been over-sampled. The main properties of the sketch proved in Lemma B.2 are about upper bounding the cost of Meyerson sketch and its size. The cost is upper bounded in the same way and the proof is intact as the sampling probabilities only increased.

The upper bound on the size can be adapted as follows. We divide the stream into chunks using the times we increased our upper bound on $M$ as pivot points. Notice that we do not need to update $L^p$

each time $M$ changes but only when $M$ increases by a constant factor. The sketch $L^{new}$ consists of the points we sample in each of these chunks. So we upper bound the size of the sketch in each chunk. The last chunk is similar to any other Meyerson sketch with a set of points being pre-selected in previous chunks. So the proof of Lemma B.2 works here as well. For any other chunk $S'$, we are running the sketch with the maximum $L^p$ at the time the points in $S'$ are arriving. So this $L^p$ is at least a constant fraction of the optimum cost of $S'$. This suffices for the argument to upper bound the size in Lemma B.2. To summarize, we have a new overall upper bound on the size of the sketch with an extra $\log(\Delta)$ as the number of chunks is related to logarithm of how much diameter grows (as we start a new chunks only when $M$ increases by a constant factor).

### F.5 Ensuring that the each sketch is computed on at most $O(w)$ points

Finally, we now show that we never need to compute any streaming algorithm on a too long stream. This can be obtained by at most doubling the time and memory requirement of the algorithm with the following technique. We keep at most 2 independent instances of our sliding window algorithm (Algorithm 2) the following way. Every $w$ steps we start a new algorithm instance that receives points from this moment on. When an algorithm instance has consumed $2w$ points that algorithm is discarded, meanwhile we update the (at most 2 active algorithms) with each point received. Any time a solution is needed we use the active algorithm that has started earlier. It is easy to see that for such algorithm all the properties needed by the theorem 3.4 are true as the algorithm has consumed the entire active window, so the solution is good. Notice that this ensures that no sketch consumes more than $2w$ points. Also notice that the running time and the memory is doubled at most.