[Reviews · NeurIPS 2020]

Review 1

Summary and Contributions: The paper considers clustering in the sliding window model. The problem is as follows: - There is a stream of points and a sliding window of size w. - At any point we wish to maintain a k-clustering of good objective value for the w last points (i.e. for the window consisting of w points) The main result of the paper is a new algorithm that maintains a constant-factor approximation at any time and uses space linear in k. I do think this is a nice result that is obtained in a quite simple way. We obtain basically a Meyerson sketch (well known technique in streaming) of two sets of data points so that the current window w is a subset of the two. To achieve this, several technicalities are needed such as running the algorithm for the possible values lambda of the current value of the optimal clustering etc. EDIT AFTER AUTHOR REBUTTAL: I read the author rebuttal and it did not change my rather favorable opinion of the paper (even though I think the argument that previous works didn't state the guarantee explicitly so we don't do either is not super convincing).

Strengths: - Natural problem. - Quite clean new algorithm that improves the space requirement both in theory and in experiments.

Weaknesses: - The experiments are run with an algorithm that is quite different from the one that is theoretically analyzed. The authors expand on this in the appendix but it worries me in that the other algorithms (especially SODA16) was not optimized in a similar way. - The constant-factor approximation is not stated. To be honest, I didn't understand why it was stated as the sketches only loses a small factor so I believe that the factor is quite good. However, when I tried to verify it, I found the Appendix in particular the proof of Theorem D.4 (lines 825-837) badly written and therefore unnecessary hard to understand what the constant in the approximation guarantee is.

Correctness: I believe they are correct. The basic idea is quite simple and valid. However, there are many details and the appendix is long so I didn't check all details.

Clarity: Main part is well written but I was disappointed with the appendix (as stated above, see also specific comments below).

Relation to Prior Work: Yes there is a clear explanation of prior work.

Reproducibility: Yes

Additional Feedback: Line 87: the word "needs" sounds like a lower bound to me. I'd use "only uses" Lemma 2.1 and elsewhere: please be specific regarding the approximation guarantee as a function of p. Half fishy to bake it in the O( ) notation As I said I took a sample of the appendix and the writing disappointed. Examples Line 827: What is m? Line 828: Well it is clear that lambda* exists. You just defined it to be (1+delta) * lambda... Line 832: from the invariants. Be more specific... Line 833: for, for Also I also found the notation that A_\tau and A_\lambda means different things quite confusing.


Review 2

Summary and Contributions: This paper introduces an algorithm to perform k-clustering on a sliding window in data streams. It uses augmented Meryerson sketches on two substreams of the data to create O(k polylog(w)) size weighted instance in each window then performs a known clustering algorithm (say ALG with an approximation factor \rho) on this instance. This results in a constant factor * \rho approximation to the clustering objective in the data window under consideration. Theoretical guarantees on the algorithms' performance are given. Extensive experiments are performed to evaluate the memory consumption, runtime, and the accuracy of this algorithm. ============================================================ Added after reading author rebuttal: I think authors have adequately addressed the concerns about the constant approximation factor in the rebuttal. I agree that those factors are rather pessimistic when compared with experimental results. Authors have promised to discuss the intuition behind Meryerson sketch and add some details of experiments on k-medians. Based on the novelty of the result and other theoretical contributions, I maintain my score.

Strengths: This paper utilizes Meryerson method which was popularly used in online setting. The algorithm overcomes the technical challenge of maintaining the weighted points only in the current sliding window (and "forgetting" old data) by using two substreams and building the sketches on them, which I think is a clever idea. The theoretical guarantees and the intuitions behind most of the steps in the algorithm are explained clearly. This paper improves the cubic dependency of space requirement on k in [17] to a linear dependency which is a significant improvement in settings where k is large. Experimental results (in the main paper and the appendices) elaborates the theoretical guarantees well. They show significant improvements in memory utilization and run-time to obtain comparable results with k-means++.

Weaknesses: The approximation factor of the clustering objective of the weighted instant constructed by Meryerson sketch is large. Even though the algorithm guarantees a constant factor, even for p=1, the approximation factor can be as large as 2^9 * the approximation factor of the used clustering algorithm. Can the authors explain how this value compares with previous results on streaming and sliding window settings?

Correctness: Although I did not check the details of all the proofs in the appendices thoroughly, the claims presented are natural and believable. The experiments compare with the previous sliding-window algorithm [17] in terms of memory utilization. Comparisons of accuracy with random-sampling baseline and k-means++ were given which I think is sufficient. I am curious to see similar experimental results for the case where p=1 (k-median) since the paper claims guarantees for p>=1 (not necessary though).

Clarity: The presentation of the paper can be improved. The intuition behind choosing the probability of adding as a new weighted point(or center) in the Meryerson sketch is not explained properly in the paper(although the exact probability is given in the appendix - it is based on the distance from the new point to the current centers). I believe this this is an important detail that needs to be explained well. Typos: A word seems to be missing in the middle of line 188.

Relation to Prior Work: Yes. This paper provides a thorough comparison with previous results' theoretical guarantees on clustering on streams and sliding windows.

Reproducibility: Yes

Additional Feedback: The ideas used in this paper are not novel by themselves but the authors use these ideas with clever modifications to overcome some technical challenges in order to construct the final algorithm and they provide a clean analysis. The theoretical guarantees are validated with detailed experiments. I think this is a nice contribution to the NeurIPS communitiy.


Review 3

Summary and Contributions: The paper presents an algorithm for k-clustering in a sliding window streaming model, where k-clustering means the generalization of k-median and k-means to any fixed l_p-norm. The main theoretical result is an algorithm that achieves O(1)-approximation for points in arbitrary metric space and thus includes the prevalent of Euclidean metric, which is also used in the experimental evaluation. This algorithm is for sliding window streaming, where the algorithm repeatedly solves the clustering problem on the w most recent points in the stream (for parameter w). While the minimal requirement is to estimate the cost of a k-clustering, this algorithm also reports k center points. The usual motivation for this model is to allow old data to expire, and analyze only recent data. As the paper mentions, expiration of old data might also be required by policies and restrictions on data retention, and therefore this model may be more valuable and timely than it seems initially. This main theorem shows that the algorithm's space complexity is about O(k (log w)^4), improving over the previous bound which grows like k^3. This should also lead to improved running time, which is more difficult to compare as it depends on running an approximation algorithm for offline. At a high level, the low space complexity follows by employing a well-known algorithm by Meyerson, which is a very simple strategy to subsample the points to something like O(k\log w), with only O(1)-factor loss in the objective. This paper views this subsample as a small sketch, because it indeed suffices to k-cluster this subsample (viewed as a weighted set). However, this approach is not applicable to the sliding window model, and the paper has to carefully manipulate the stream before applying Meyerson's sketch (polylog(w)-many times), in an ingenious manner. I should note that standard methods for the sliding window model (like smooth histograms) are not applicable here. In this sense, the paper really solves a difficult problem. ADDED AFTER AUTHOR REBUTTAL. I understand the clarifications. My evaluation has not changed.

Strengths: The paper solves a difficult theoretical problem, using new ideas. The results are applicable to a broad range of k-clustering problems, including different objectives (e.g., k-median and k-means) and every metric space (including Euclidean) The experimental evaluation shows that this new algorithm is quite efficient in comparison with the previous algorithm and other baseline solutions, and yields solutions with low cost (objective function) The sliding window model may be more valuable and timely than it seems initially.

Weaknesses: The theoretical guarantee is O(1)-approximation, which could be a large constant, and not say 1+epsilon or even a small explicit constant like 2

Correctness: Seem correct, but part of the analysis is deferred to the supp material, which I did not read.

Clarity: The paper is generally well written, although it's not clear in the first few pages whether the setting is Euclidean or a general metric (and how is that metric accessed). The plots are very small and thus hard to read

Relation to Prior Work: The prior work is discussed and compared to.

Reproducibility: Yes

Additional Feedback: L 41-31: not clear how you chose these references, are these the first or latest word? Euclidean or metric? L 125: the big O must hide a term depending on \alpha, hence it's better to write it explicitly L 130: The NP-hardness does not imply any lower bounds on the space complexity, right? L 185: \hat f was not defined Algorithm 1: why is it called "Update of Meyerson"? This is the entire algorithm (not even described in a streaming fashion of item-by-item) Table 1: why measure the last column in percentage? for example, is it 102% improved cost or worse cost? why not the ratio/factor between costs is 2.02? L 287: cost means time? Table 2: Max percentage is out of w? Why not list also a comparison of the cost (objective value)? L 301 and 326: which of the baseline algorithms? you mentioned a few, so perhaps you should name them L 325: I don't see that W grows in table 2


Review 4

Summary and Contributions: I would like to mention that I have reviewed this paper in the past (submitted to another venue). - Consider following definitions: - k-clustering problems (k-means/median/center) problems are well known. - Sketching is an algorithmic technique where a small summary of the input Data is maintained for approximation some specific property of the data. - Sliding window algorithm is an algorithm in the streaming model such that the algorithm gives guarantee for the last w items of data seen in the stream. - The paper gives sketching algorithm in the sliding window model. Previous known results gave either sketching algorithm for the entire dataset seen so far in the stream or gave algorithm with worse dependency on the space requirement (k^3 versus k). Here are some comments about the writeup: 1. Lines 60-71: It will be nice to also have the comparison with previous work with respect to running time in this paragraph. 2. You maintain two sketches during the execution of the algorithm that suffices for finding a good set of centers for the sliding window. Do you think maintaining coresets similarly might also work? Coresets are powerful objects in the context of k-means/median clustering. It may be worthwhile adding a discussion in case you have given this some thought. 3. Any comment/discussion on the tightness of the approximation bounds obtained would be nice even though I understand that obtaining the tightest possible approximation ratio is not the main agenda of this work. 4. It may be better to state clearly what m and M are in Lemma 3.1. 5. Line 186: "Note that when M and ...". Did you mean M/m instead of M? I have read author rebuttal. There is no change in my review post rebuttal.

Strengths: - Meyerson’s sketching technique is a simple and practical algorithm in the context of k-clustering problems. The paper extends this to the sliding window model. People interested in using sketching in the sliding window model should find this interesting.

Weaknesses: There are settings where improvement from k^3 to k is significant. I am not sure if this improvement is interesting in most settings.

Correctness: Yes

Clarity: Yes. There has been improvements over the previous version.

Relation to Prior Work: Yes

Reproducibility: Yes

Additional Feedback:

[Author Response · NeurIPS 2020]

We thank all the reviewers and ACs for their work in this challenging time. We will fix all typos found and improve the
presentation of the appendix.

**Constants in the approximation factor**    We agree with the reviewers that the constants in the statements of our
theorems are not small. We stress that as in prior work (see [17], and [42]) we did not optimize for the worst case
approximation constants. The bulk of the constant comes from the use of the well-known Meyerson sketch. Our
results are very close to that of the simpler insertion-only case. Specifically, [42] shows a $2^{3p+5}\rho$ approximation for the
insertion-only case (here $p$ is the norm, $\rho$ is the offline algorithm factor). This compares with our result in Lemma D.1
which is only a factor of 2 higher, $2^{3p+6}\rho$, but solves the more general problem. Note that the the SODA16 baseline [17]
does not explicitly state the constant factor thus making a precise comparison difficult, but it uses the same Meyerson
sketch as a subroutine thus incurring similar constants (see Lemma 3.2 in [17]). Importantly, these large factors appear
only in the worst case analysis, and as our experiments (as well those of [42]) confirm that real instances behave much
better. This could be explained by the fact that only a small fraction of all orderings of a set of points lead to large
approximations in the Meyerson sketch (this has been shown formally by prior work).

**Reviewer 1:**    For our algorithm we used a standard optimization from the literature ([42]): running one copy of the
Meyerson sketch instead of the $O(\log(1/\gamma))$ copies that are needed for high probability statements. We also developed
a lazy evaluation of the cost of the Meyerson sketch that saves update time. Notice that for fairness of comparison
with SODA16 [17] (which uses at its core a Meyerson sketch as well) we use the same implementation of the sketch
(with the same optimizations) for the two algorithms. This shows that our speedups over SODA16 come mostly from
avoiding the additional factors of their algorithm; we will clarify this detail in the updated paper. We will also clarify
in the $O()$ notation the dependency on $p$. We will improve the presentation in the appendix and we will change the
notation of $A_\tau$ vs $A_\lambda$ as suggested.   *Q: Line 827: What is $m$?*   A: It is the lower bound on the optimum as in the
preliminary section. We will clarify that here we mean that, for the first threshold in $\Lambda$, the associated sketch $A_m$ is not
empty.   *Q: Line 828: Well it is clear that $\lambda*$ exists.*   A: We mean that it is in the set of the thresholds $\Lambda$ for which we
computed a sketch.   *Q: Line 832: from the invariants.*    A: We will clarify that it is invariant (iv) in Lemma D.2

**Reviewer 2:**    Please see the answer above on the approximation factor. We will state the intuition behind the probability
of adding a center to the Meryerson sketch. This is now a standard approach; the reason is that a point that is far from
the current centers should be be added to avoid a large cost for that point. We ran experiments with k-median objective,
$p = 1$; the results are in line with that of $p = 2$. For instance, for vmeasure accuracy over our datasets with ground-truth
(see L.327 for the setup) the offline algorithm, our stream algorithm and the sampling baseline obtain $81.8\%$, $79.9\%$,
$78.5\%$, respectively, confirming the same trend observed for k-means. We will add more details in the paper.

**Reviewer 3:**    We will clarify at the beginning of the paper that the algorithm works on arbitrary metric spaces which
we access only through distance function evaluations. We will increase the size of the plots.   *Q: L 41-31 .. Euclidean
or metric?*    A: We will clarify the citations. k-median, k-means, and k-center are NP-Hard even in the Euclidean
case but there are constant factor approximation algorithms for the general metric space case as well.    *Q: Lower
bounds on the space complexity*   A: Correct, but it is trivial to show that at least $k$ points must be stored to provide
any approximation.   *Q: L 185: $\widehat{f}$ was not defined*   A: We apologize for the notation, $\widehat{f}$ is defined in that line as an
approximation to the cost of the $\epsilon$-consistent mapping that is computed by the algorithm   *Q: Algorithm 1: why is it
called "Update of Meyerson"?*    A: we will rename it as it is confusing, it processes indeed the entire stream not a
single update.   *Q: Table 1: why not use ratios?*   A: We will use ratios to make the table easier to read. We reported
the ratio of the cost of our algorithm over the baseline as a percentage, i.e. $102\%$ means that the cost is $1.02$ times the
baseline cost.   *Q: L 287: cost means time?*   Correct, we mean update time.   *Q: Table 2: Max percentage is out of $w$?*
A: Yes   *Q: Why not list also a comparison of the cost (objective value)?*    A: For lack of space it is in supplemental
material, Table 7, L.939   *Q: L 301 and 326: which of the baseline algorithms?*   A: We will clarify better, in L.301 we
are evaluating SODA16 while in L.326 we are comparing with the offline K-Means++ baseline.   *Q: L 325: W grows
in table 2*   A: We apologize. This is visible in the supplemental material Table 5, L.923.

**Reviewer 4:**    Concerning the settings where the improvement from $k^3$ to $k$ is significant, we would like to stress that
in many industrial applications on large scale datasets the number of clusters can be quite large. We provided some
examples in the introduction. Empirically we observe that that our speedups over the prior SODA16 work are an order
of magnitude even for $k$ as low as $4$ in the COVERTYPE dataset. Moreover, our algorithm is significantly simpler
than the previous one.   *Q: 1. Lines 60-71*   We will discuss the running time in the related work.   *Q: what $m$ and $M$
are in Lemma 3.1.*   We will clarify that they are the lower and upper bound on the cost of the optimum as defined in
the preliminaries.   *Q: Line 186: "Note that when M and ...". Did you mean M/m instead of M?*   Correct, our bound
depends on $M/m$ being polynomial in $w$.

[Meta-Review · NeurIPS 2020]

The paper considers k clustering problem in the sliding window streaming model. The result is nice. However, the authors are urged to consider the reviews when preparing the final version including explicitly stating a bound on the constant factor approximation obtained.